# Touchless interactive teaching of soft robots through flexible bimodal sensory interfaces

Wenbo Liu [1,7], Youning Duo[1,7], Jiaqi Liu [1,7], Feiyang Yuan[1,7], Lei Li [1], Luchen Li[1], Gang Wang[1], Bohan Chen[1], Siqi Wang[1], Hui Yang[2], Yuchen Liu[3], Yanru Mo[3], Yun Wang[1], Bin Fang[4], Fuchun Sun[4], Xilun Ding [1], Chi Zhang [5,6] & Li Wen [1] ✉

In this paper, we propose a multimodal flexible sensory interface for interactively teaching soft robots to perform skilled locomotion using bare human hands. First, we develop a flexible bimodal smart skin (FBSS) based on triboelectric nanogenerator and liquid metal sensing that can perform simultaneous tactile and touchless sensing and distinguish these two modes in real time. With the FBSS, soft robots can react on their own to tactile and touchless stimuli. We then propose a distance control method that enabled humans to teach soft robots movements via bare hand-eye coordination. The results showed that participants can effectively teach a self-reacting soft continuum manipulator complex motions in three-dimensional space through a "shifting sensors and teaching" method within just a few minutes. The soft manipulator can repeat the human-taught motions and replay them at different speeds. Finally, we demonstrate that humans can easily teach the soft manipulator to complete specific tasks such as completing a pen-and-paper maze, taking a throat swab, and crossing a barrier to grasp an object. We envision that this user-friendly, non-programmable teaching method based on flexible multimodal sensory interfaces could broadly expand the domains in which humans interact with and utilize soft robots.

Soft robots have attracted growing attention for their enormous potential in real-world applications[1–8]. Because they are highly conformable, soft robots have extraordinary advantages over rigid robots for safely interacting with humans in a wide range of environments[9–13]. However, because soft robots are challenging to model and program, non-specialists often face non-negligible obstacles when working with soft robots to achieve specific movements and perform certain tasks[14–18]. An interactive teaching method, which could efficiently and flexibly "teach" soft robots movement patterns, would dramatically benefit human users at home, on production lines, and in other unstructured environments (Fig. 1). Unlike rigid robots[19–21], there are very few studies demonstrating the teaching of soft robots through human interaction. This is because there are two primary challenges to achieving soft robotic teaching through human interaction: the process requires (1) a multimodal, versatile, and robust flexible sensing device for interactions between a soft robot and human demonstrator; and (2) a user-friendly, non-programmable teaching method to transfer a human demonstrator's instructions to the soft robots.

Regarding the first challenge, most previous studies have focused on tactile sensing for soft robots that can only respond to physical

[1]School of Mechanical Engineering and Automation, Beihang University, Beijing 100191, China. [2]Institute of Semiconductors, Guangdong Academy of Sciences, Guangdong 510075, China. [3]School of General Engineering, Beihang University, Beijing 100191, China. [4]Tsinghua National Laboratory for Information Science and Technology, Department of Computer Science and Technology, Tsinghua University, Beijing 100084, China. [5]CAS Center for Excellence in Nanoscience, Beijing Key Laboratory of Micro-Nano Energy and Sensor, Beijing Institute of Nanoenergy and Nanosystems, Chinese Academy of Sciences, Beijing 101400, China. [6]School of Nanoscience and Technology, University of Chinese Academy of Sciences, Beijing 100049, China. [7]These authors contributed equally: Wenbo Liu, Youning Duo, Jiaqi Liu, Feiyang Yuan. ✉e-mail: liwen@buaa.edu.cn

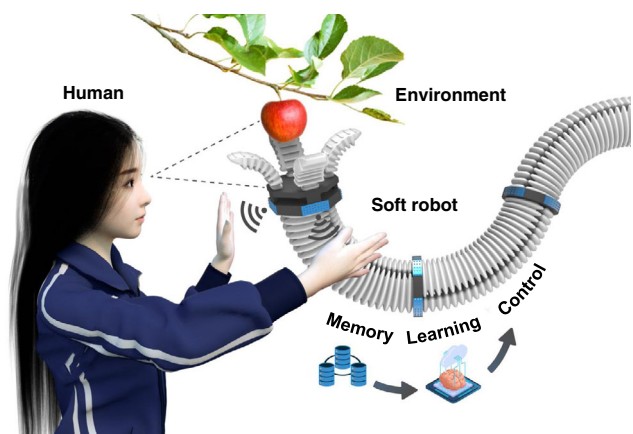

**Fig. 1 | A depiction of the principle of touchless human-soft robot interaction based on flexible smart skin.** Humans can "teach" the robot to accomplish various tasks by manipulating the robot in a touchless manner.

touch and not respond to touchless stimuli. Triboelectric nanogenerator (TENG), which harnesses the coupled effect of contact electrification and electrostatic induction, can transduce both tactile and touchless stimuli into electrical signals[22–26]. Triboelectric sensors based on TENGs have unique advantages for soft robots due to their wide-ranging material makeup (e.g., from low to high Young's modulus), easily fabricated simple structure, high sensitivity, and fast response times[27–32]. Previous studies that utilized flexible triboelectric materials and structures have made remarkable progress in pressure and stress sensing[33–38]. Preliminary works that explore touchless sensing have also emerged[39–42]. However, because tactile and touchless stimulations result in identical trends in electric variation, it is challenging for triboelectric sensors to distinguish between tactile and touchless signals accurately in real-time[43,44] (Supplementary Fig. 1A, B and Supplementary Movie 1). Thus, flexible triboelectric sensors capable of tactile and touchless real-time sensing remain to be researched, which may lay a research foundation for a new paradigm of soft robotic interactive teaching.

Regarding the second challenge, the interactive teaching of soft robots (e.g., the soft continuum manipulator) is little understood. Traditionally, the primary principle has been based on contact teaching for rigid robotic manipulators with few degrees of freedom[45,46]. This principle was commonly achieved by manually moving the manipulators under controlled, low-impedance modes while the manipulators' encoders recorded the kinematics of the teaching process for replaying the motion. However, this type of contact teaching cannot be applied to soft robots for two reasons. First, the infinite degrees of freedom and compliant nature of a soft continuum manipulator make it challenging for a user to control explicitly, unlike the discrete configurations of a rigid manipulator[47]. Second, the contact-based teaching method for soft continuum robots produces passive deformation, and measuring these deformed configurations requires a large number of soft sensors (either embedded in or on the surface of the robot) to reconstruct the robot's three-dimensional kinematics[48,49]. Given these challenges, is it possible to interactively teach soft robots through a flexible sensory interface? Can non-specialist users instruct soft robots to realize operational tasks in unstructured environments without programming?

Here, we develop a flexible bimodal smart skin (FBSS) with both tactile and touchless sensing by integrating a triboelectric sensor with a liquid metal sensor. The triboelectric sensor can respond to touchless stimulation, and the liquid metal sensor can respond to tactile stimulation. On this basis, the implemented FBSS can unambiguously distinguish between tactile and touchless modes in real time. We then characterize the sensing performance of the FBSS for both tactile and touchless sensing. Finally, we build a control framework for interactive teaching with the FBSS. We also propose a "shifting sensors and teaching" method for teaching complex locomotion, which involves moving FBSS to different locations on a manipulator during a teaching session. We show that a non-specialist can efficiently and interactively teach a continuum soft manipulator picking-and-placing, painting, throat-swabbing, and crossing a barrier to grasp an object. In addition, we also test the interactive performance of the FBSS on other soft robots including a soft origami robot and a robotic gripper.

## Results

### Working principle and sensing performance of FBSS

The flexible bimodal smart skin (FBSS) structure contains five flexible layers (Fig. 2a and "Methods" section). The flexible dielectric layer was fabricated by casting silicone rubber (Smooth-on, Dragon skin 00-20) in the mold with pyramid-shaped microstructures. The flexible electrode layer was fabricated with patterned Ag nanowire (NW) networks and was transferred by mixing polydimethylsiloxane (PDMS) base with a curing agent (Dow Corning, Sylgard184) at a typical weight ratio of 10:1. The stimulation layer, underneath the electrode layer, was fabricated using a similar method to that of the flexible dielectric layer. The surfaces of the flexible dielectric layer, flexible electrode layer, and stimulation layer formed chemical bonds after being treated with plasma (OPS plasma, CY-DT01). The liquid metal layer was first printed with a liquid metal printer (DREAM Ink, DP-1) and then the package layer (Smooth-on, Dragon skin 00-20) was used to transfer and contain the liquid metal. The stimulation layer was bonded to the package layer via a silicone rubber adhesive (Smooth-on, Sil-Poxy). The electron microscope image was taken for the fabricated pyramid-shaped microstructures, the height and width of the pyramid-shaped microstructures are 320 μm and 500 μm, respectively (Fig. 2b). The optical photo was taken for the printed liquid metal pattern, and the width of the liquid metal line is about 300 μm (Fig. 2c). The FBSS can be folded and stretched (maximum stretching rate is 58.4%), which demonstrates its excellent flexibility and stretchability (Fig. 2d, e).

The complete tactile and touchless perception principle of the FBSS is divided into multiple stages (Fig. 2f). During the initial stage (i), equal negative and positive charges are generated on the flexible dielectric layer and external object from different electron affinities after a few repeated contacts. These surface charges can remain for a sufficient time (over 1 h) for the interactive teaching process (Supplementary Fig. 2). At stage (ii), as the external object approaches the flexible dielectric layer, the electric potential between the electrodes and ground will be changed, which drive free electrons to flow from the ground to the flexible electrode, thus generating a current in the circuit. Note that the resistance of the liquid metal sensor remains stable as no contact pressure force acts on the FBSS during this stage. In stage (iii), the FBSS starts to deform with contact pressure from the external force that acts on the silicone rubber. The external object is closer to the flexible electrode during this stage, so the free electrons flow further from the ground to the flexible electrode and generate a current in the same direction. The liquid metal layer is compressed and the cross-sectional area of the liquid metal channel decreases, causing its resistance starts to increase. During stage (iv), when the external object is entirely in contact with the FBSS, the distance between the object and the flexible dielectric layer is compressed to the minimum. Charge neutralization occurs and the free electrons stop moving and the resistance of the liquid metal reaches the maximum. In stage (v), when external pressure is released, the free electrons flow back from the flexible electrode to the ground and generate a current in the opposite direction. The resistance of the liquid metal decreases with the recovery of the channel shape. In stage (vi), when the external object separates from the flexible dielectric layer, the number of electrons increases that flowing back to the ground, and generate a current in the

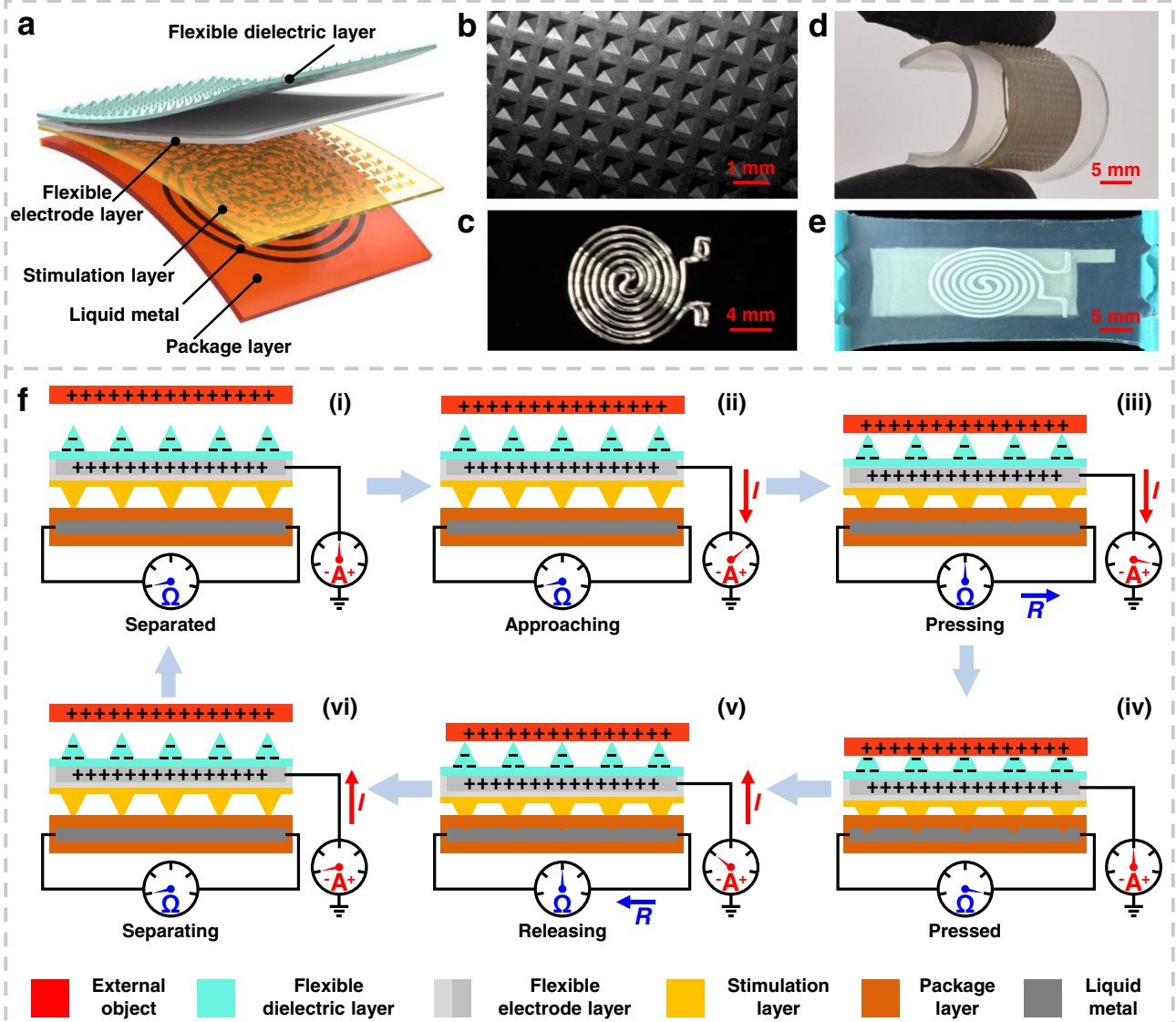

**Fig. 2 | Design and sensing mechanism of the proposed tactile/touchless flexible bimodal smart skin (FBSS). a** Soft sensor design with different functional layers stacked together. These layers include a flexible dielectric layer (cyan), a flexible electrode layer (gray), a stimulation layer (pale yellow), the liquid metal material (black), and a package layer (orange). **b** Electron microscope image of the micro-pyramid structures on the top side of the flexible dielectric layer. **c** An optical microscope image of the liquid metal material printed on the silicone material layer. **d** A bending photograph of the FBSS prototype demonstrates its flexibility. **e** A stretching photograph (maximum stretching rate is 58.4%) of the prototype demonstrates its stretching ability. **f** The tactile/touchless sensing mechanism of the prototype: (i) The equal density of negative and positive charges was generated on the flexible dielectric layer (gray) and external object (red) due to the different electron affinities after a few contacts. (ii) The free electrons were driven to flow from the ground to the flexible electrode as the external object approached the flexible dielectric layer. (iii) The external object (red) starts to contact the FBSS, increasing the transfer of electrons, and the liquid metal resistance increases. (iv) The external object (red) is entirely in contact with the FBSS; charge neutralization occurs and the free electrons stop moving and the resistance of the liquid metal reaches the maximum. (v) As the external pressure was released, the electrons flowed back from the flexible electrode (gray) to the ground, and the resistance of the liquid metal decreased as the channel recovered to its initial state. (vi) As the external object (red) was separated from the FBSS, the backflow electrons increased and the liquid metal's resistance remains stable.

same direction as the previous state. The resistance of the liquid metal remains stable as the disappearance of physical contact between the external object and FBSS. Finally, when the external object is far away from the FBSS, a new electrical equilibrium is established.

We implemented a measurement system to investigate the performance of the FBSS (Supplementary Fig. 3). The FBSS was fixed on a flat plate assembled on top of a force gauge (ATI Industrial Automation, mini40). The external object was attached to the end of the linear motor actuator (LinMot, E1100), which can cyclically approach and press the FBSS. A piece of glass, set 20 mm away from the FBSS, was used as the external object for the tactile and touchless sensing tests. The effect of approach distance on the output signals of the FBSS was

first tested (Fig. 3a). The touchless output signal $\Delta U$ decreased exponentially from 11.35 to 0 V as the distance increased from 0 to 20 mm. The tactile output signal $\Delta R$ of the FBSS remained stable without variation. Electrostatic induction steadily weakened as the distance increased between the external object and the FBSS, and the output voltage decreased in step. We also studied the relationship between output signals and vertical pressure acting on the FBSS (Fig. 3b). As pressure increased from 0 to 30 kPa, the tactile output signal $\Delta R$ of the FBSS increased from 0 to 17.24 Ω and the touchless signal $\Delta U$ increased from 0 to 3.2 V. The cross-sectional area of the liquid metal channel decreased with increasing external pressure, which resulted in its resistance increase. With the pressure increase, the external object is

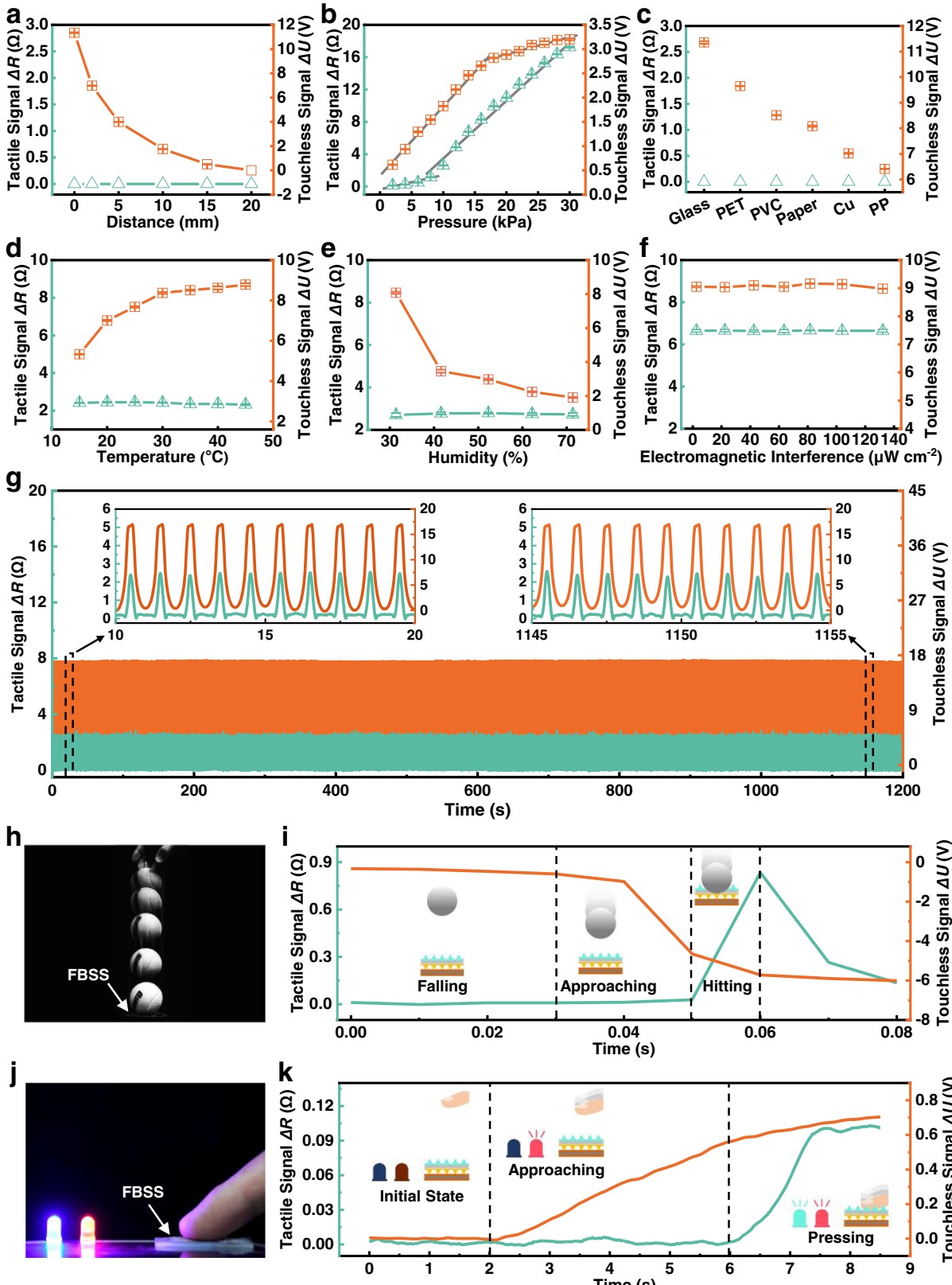

closer to the FBSS. This also enhances the electrostatic induction between the external object and the FBSS. Because different materials have different electron affinities, material types can affect the surface charge density of the flexible dielectric layer. Therefore, the FBSS can be used for material identification (Fig. 3c). The tactile signal $\Delta R$ always remained at 0 kPa, while the touchless signal varied per material with a

test distance of 20 mm. This allows the FBSS to distinguish between materials in real time. The dynamic response of the tactile sensing of the FBSS is about 120 ms, which is close to that of human skin (Supplementary Fig. 4A, B). The tactile and touchless signal noises are 0.04 Ω and 0.12 V, respectively (Supplementary Fig. 5A, B). The maximum signal-to-noise ratio (SNR) of the touchless signal and tactile

**Fig. 3 | Characterization results of the proposed FBSS prototype for tactile and touchless sensing. a** Tactile (cyan) and touchless (orange) output signals were tested under different distances between a surface (glass) and the FBSS. **b** Tactile and touchless output signals under different loading pressure. The supplementary materials and methods provided more details about the loading experiments. **c** FBSS's output signals at surfaces with different materials (with an above distance of 20 mm). The reactions of the FBSS to environmental change, including **d** temperature, **e** humidity, and **f** electromagnetic interference. **g** The prototype's stability and durability tests show that the FBSS can sustain over 1200 loading-unloading cycles of compression tests with a distance of 20 mm and a pressure of 10 kPa. **h, i** High-speed images and output signals of the FBSS when a tennis ball drops onto it (drop distance: 200 mm). **j, k** Image and output signals of the FBSS when a human finger presses on it. A red light-emitting diode (LED) was programmed to turn on when the induced touchless signal exceeded a threshold value; the blue LED turned on when the finger touched the FBSS. Error bars represent standard deviation, $n = 5$ independent replicates.

signal are 94.58 and 431.03, respectively (Supplementary Fig. 5C, D). The maximum resolutions measured in the touchless and tactile experiments are 0.05 mm and 0.35 kPa, respectively (Supplementary Fig. 6A, B).

To evaluate how environmental factors affect the sensing performance of the FBSS, we experimentally tested the effects of temperature, humidity, and electromagnetic interference on the FBSS. The output touchless signal increases as the temperature increases from 15 to 30 °C, and then remains stable with further temperature increases (Fig. 3d). The output tactile signal remains almost invariant with an increase in temperature. We investigated the effect of humidity on the output signals of the FBSS (Fig. 3e). The touchless signal decreases gradually as humidity increases from 31.4 to 71.4%. The output tactile signal remains almost invariant with an increase in humidity. The touchless and tactile signals remain unchanged with an increase in electromagnetic interference (Fig. 3f). The long-term stability of the FBSS is also validated under an external pressure of 10 kPa and a distance of 20 mm. We measured outputs of the FBSS over 1200 cycles in the same condition (Fig. 3g). The results show no obvious waveform changes, which points to the long-term usage of the FBSS.

We performed a series of tests to verify the FBSS's sensing ability when interacting with humans and the external environment. Firstly, the FBSS was used to detect the falling process of a tennis ball, which was placed above the FBSS at an initial height of 200 mm (Fig. 3h). A high-speed camera (Photron Ltd, FASTCAM Mini UX100) recorded the entire process at a sampling rate of 250 fps, while the FBSS recorded both tactile and touchless signals. The falling process was divided into three stages (Fig. 3i and Supplementary Movie 2). In stage (i), the tennis ball starts to fall from the initial height and approaches the FBSS. However, the tennis ball has not yet entered the detection range of the FBSS, so its output signals remain stable. In stage (ii), the tennis ball continues to fall and enters the FBSS's detection range. The touchless signal $\Delta U$ decreases from 0 to −4.56 V and the tactile signal $\Delta R$ remains 0 Ω because the tennis ball has not yet come into contact with the FBSS. In stage (iii), the tennis ball contacts the FBSS. The touchless signal $\Delta U$ decreases further from −4.56 to −5.81 V and the tactile signal $\Delta R$ drastically increases from 0 to 0.83 Ω. In addition, we also show that the FBSS can perceive and distinguish the touchless distance of a feather falling through the air (Supplementary Movie 3).

We tested the tactile and touchless sensing ability of the FBSS on a human finger. The FBSS was connected to a sample circuit that controls two LEDs based on tactile (blue LED) and touchless (red LED) sensory feedback (Fig. 3j). We recorded the entire process of a finger approaching and pressing the FBSS (Fig. 3k and Supplementary Movie 4). During stage (i), the finger was 50 mm away from the FBSS's surface, both LEDs were off. During stage (ii), as the finger approached the FBSS, the red LED lit up while the blue LED remained off. The recorded sensory data showed that the output touchless signal $\Delta U$ increased from 0 to 0.56 V while the output tactile signal $\Delta R$ remained unchanged. During stage (iii), the finger pressed on the FBSS, the blue LED lit up and the red LED's brightness increased. This result intuitively demonstrates that the FBSS can perceive tactile and touchless information from a human finger.

## Self-reacting soft robots equipped with FBSS

To equip a soft robot with the FBSS, we integrated the FBSS with a soft manipulator segment that could be bent and shortened. A human hand could touchlessly control the soft manipulator segment's bending and shortening motions (Fig. 4a, b and Supplementary Movie 5). The soft manipulator segment was programmed to deform when the touchless output signal of the FBSS reached a predetermined threshold value (the control flowchart is provided in Supplementary Table 1). A soft origami robot equipped with an FBSS buried in the sand could perceive the approach of a robotic bug and grasp the bug by inflating its actuator (Fig. 4c and Supplementary Movie 6).

By integrating the FBSS with the tip of a soft robotic gripper, we endowed it with the ability to "search and grasp" objects through tactile and touchless sensing (Supplementary Fig. 7A, B and Supplementary Movie 7). The whole process can be divided into different stages (Fig. 4d, e). Initially, both the tactile signal $\Delta R$ and the touchless signal $\Delta U$ were negligible. As the rigid robotic arm moves horizontally and the gripper approaches the plastic cylinder, the touchless signal $\Delta U$ starts to rise and the tactile signal $\Delta R$ remains low. The threshold for the touchless signal to "identify" a target object was set to 0.1 V. Once the signal surpassed this level, the soft robot began to grip the target. The tactile signal $\Delta R$ rose and the touchless signal $\Delta U$ increased further until a stable grip was achieved. These experimental scenarios illustrate that the FBSS can effectively enable soft robotic interactions through tactile and touchless perception.

## Interactive teaching of the soft manipulator

To further explore the more intelligent interaction between the soft robot and humans, we presented a flexible interface and interactive method with the FBSS (Fig. 5 and "Methods" section). Through the flexible interface and interactive method, we demonstrated that humans could interactively teach a soft manipulator to move in two-dimensional (2D) and three-dimensional (3D) space.

Based on the interactive teaching method, a user taught the soft manipulator to grasp an object in 2D space (Fig. 6a and Supplementary Movie 8). For a simpler explanation, we divide the teaching process into four steps. In step (i), we showed the user's ability to control the initial length of the soft manipulator by altering the distance between the user's hand and the FBSS. This step allowed the user to select an effective length of the soft manipulator in the first 5 s. In step (ii), the user touchlessly "bent" the soft manipulator by approaching the FBSS sensor with a hand. More specifically, the user chose to apply a strategy of multiple approaching-leaving actions to "bend" the soft manipulator in several large, discrete steps to move it towards the target object. During this process, the fluctuation range of the normalized touchless signal $\Delta U$ varied from 0 to almost 1. In contrast, the output tactile signal $\Delta R$ remained unchanged at nearly 0. In step (iii), when the soft manipulator approached the target position, the user switched from large steps to small steps to move the soft manipulator to the final few centimeters. Here, the normalized touchless sensory output fluctuated between 0.2 and 0.5. In step (iv), when the soft gripper reached the target object, the user pressed the FBSS to trigger the grasping motion. The drastic increases in the normalized touchless and tactile signals can be observed. Through our logic algorithm, the soft manipulator terminates the teaching task and closes the gripper when the tactile

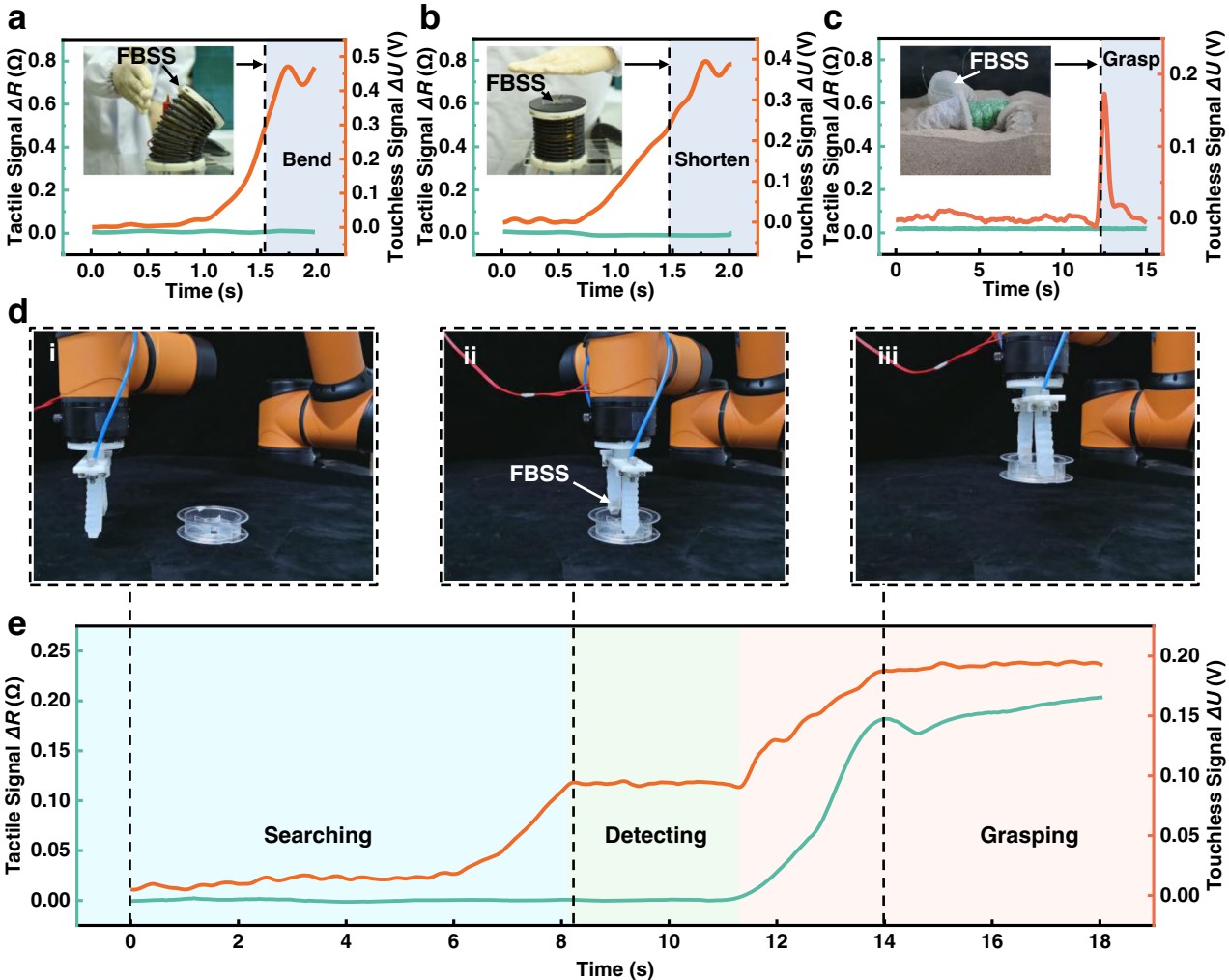

**Fig. 4 | Self-reacting soft robots triggered by touchless external stimuli from human and environment.** Touchless control of a pneumatic-actuated 3 DoFs soft manipulator for **a** bending and **b** shortening in response to an approaching human hand. **c** A pneumatic-actuated soft origami robot with the FBSS successfully detects and grasps a toy bug. **d** Demonstrating a soft robotic gripper equipped with the FBSS autonomously searching, detecting, and grasping a plastic cylinder object. **e** The tactile and touchless sensory outputs during this process are plotted against time. The searching (cyan shading), detecting (green shading) and grasping (orange shading) processes can be distinguished in real time.

signal output exceeds a threshold value (Supplementary Table 2). Finally, the soft manipulator successfully gripped the target object and automatically returned to its initial position.

As the system simultaneously records the drive step size sequence of the soft manipulator in the teaching process, one can actuate the soft manipulator to repeat the movements and "replay" the entire taught motions. We compared the real-time driven air pressure during the teaching and repeating processes (Supplementary Fig. 8A, B). The two air pressure curves show almost identical changes during both processes. We demonstrated the interactive teaching results by showing the soft manipulator grasping objects at low, medium, and high positions (Supplementary Movie 9). The teaching process took 53, 53, and 59 s, respectively. The trajectories of the soft manipulator during the teaching and replaying phases coincided well with each other (Fig. 6b–d). One can also replay the teaching trajectories in a sped-up and slowed-down manner (Supplementary Movie 10), adding to the flexibility of the manipulator's task execution.

We also performed interactive teaching of object-grasping in a constrained environment, where an obstacle was placed on the path of the soft manipulator (Supplementary Fig. 9A–E). The results show that the soft manipulator can successfully grasp an object within 40 s while encountering an obstacle that causes contact deformation

(Supplementary Movie 11). In typical situations, enabling soft manipulators to work in a constrained environment requires a great deal of modeling and programming work. In contrast, no additional programming was required with the current interactive method.

We realized interactive teaching in 3D space by integrating two FBSSs on the soft manipulator (The logic algorithm is shown in Supplementary Table 3). The user interactively taught the soft manipulator to grasp an object out of the bending plane with both hands (Fig. 6e and Supplementary Movies 12, 13). The entire teaching process can be divided into four steps. In step (i), as with interactive teaching in planar space, the user applied a left hand to "bend" the soft manipulator in several large, discrete steps via multiple approaching-leaving actions (applied to FBSS I). The normalized touchless signal $\Delta U$ of FBSS I ranged from 0 to about 1. In step (ii), the user switched from a large to small step size to move the soft manipulator more slowly and position the soft gripper on the same horizontal level as the target object. The normalized touchless signal $\Delta U$ of FBSS I remained around 0.5 during this stage. In step (iii), the user applied a right hand to FBSS II, moving the soft manipulator out of the original plane. The end effector reached the target object after a few of these repeated right-hand approaching-leaving actions. The normalized touchless signal $\Delta U$ of FBSS II also ranged from 0 to about 1. Finally, in step (iv), once the soft gripper

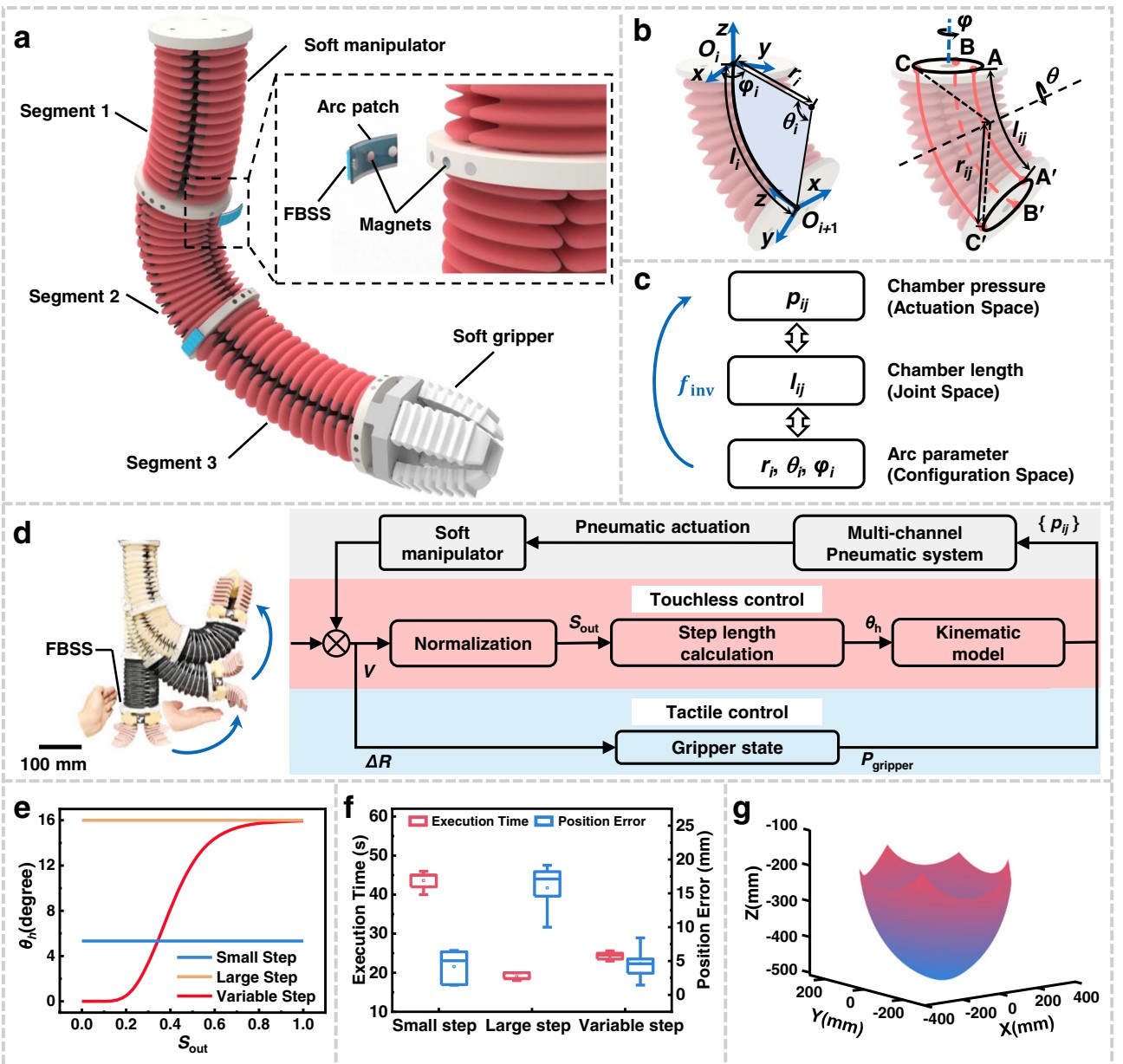

**Fig. 5 | Interactive teaching kinematics control method for the soft manipulator. a** A schematic view of the soft manipulator, consisting of three segments, each containing three chambers that actuate pneumatically. The FBSS is placed on a flexible, arc-shaped patch with three magnets on the back. Small magnets were also placed around the bottom of each segment of the soft manipulator, so the position of the FBSS can be quickly shifted. **b** Geometric functions in the bending segment, where $\varphi_i$ is the segment deflection angle around the z-axis; $\theta_i$ is the segment curvature angle around the y-axis; $r_i$ is the segment curvature radius; $l_i$ is the arc length of the segment; $r_{ij}$ is the curvature radius of each actuator; $l_{ij}$ is the arc length of each actuator. **c** The actuation, joint, and configuration spaces and mapping between them, define the inverse kinematics ($f_{inv}$). **d** The closed-loop control framework for interactive teaching, where $V$ represents the measured voltage of the FBSS, $S_{out}$ is the normalized voltage signal, $\theta_h$ is the calculated step length, and $p_{ij}$ represents the pneumatic pressure of each soft actuator. **e** The mapping of the step length $\theta_h$ and normalized voltage signal $S_{out}$. **f** Interactive teaching experimental results: execution time and the position error of the soft manipulator as a function of step length. Box plots indicate median (middle line), 25th, 75th percentile (box) and maximum and minimum values (whiskers) as well as average values (single points). **g** The simulated workspace of the soft manipulator with interactive teaching. Error bars represent standard deviation, $n = 5$ independent replicates.

reached the target object, the user pressed the FBSS I to trigger grasping. We compared the real-time driven air pressure during both the teaching and repeating processes (Supplementary Fig. 10A, B). The two air pressure curves show almost identical changes during both processes. We also demonstrated that interactive teaching allows the soft manipulator to grasp objects in low, medium, and high positions, respectively (Supplementary Movie 14). These teaching processes took 51, 56, and 61 s, respectively. All three manipulator trajectories during teaching and repeating show excellent agreement (Fig. 6f–h).

This result indicates that users can effectively teach the manipulator to move and perform actions in 3D space. Since the experiment, more than ten novices have been successful at interactively teaching the soft manipulator to grasp a target object in 3D space.

To demonstrate an intelligent FBSS placement strategy, we used an FBSS to control multiple motion modes for the soft manipulator. By changing the mounting position of the FBSS on the soft manipulator, it can be taught to move left, right, and backward (Supplementary Fig. 11 and Supplementary Movie 15). The trajectories of the soft manipulator

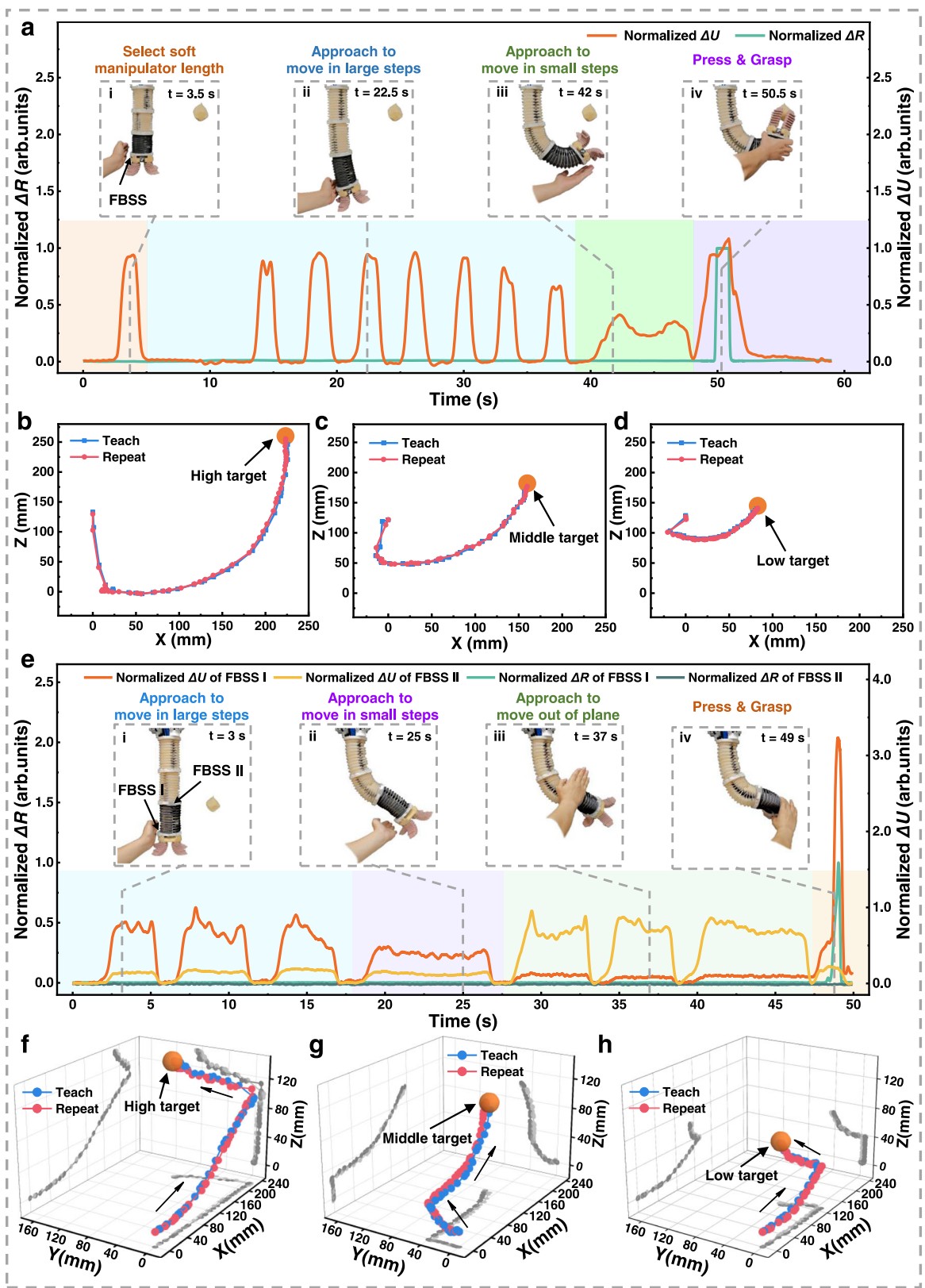

during the teaching and replaying phases, as in the previous experimental procedure, coincided well with each other. To evaluate the usability of the interactive teaching method, we performed a teaching experiment with multiple participants, including two sophisticated experts (researchers of this project) and three novices with no experience with an interactive robotic system. We attached a laser pointer to the end of the soft manipulator to assess positioning

accuracy (Supplementary Fig. 12A). The participants could control the position of the laser pointer on a target by touchlessly teaching the soft manipulator. We measured the positioning error after each participant's teaching sessions ten times. The results showed that the positioning error for the experts was <10 mm for all ten trials, while the positioning error for the novices was relatively large in early trials (Supplementary Fig. 12B). Notably, after at most 8 attempts, all the

**Fig. 6 | Interactive teaching of the self-reacting soft manipulator in 2D and 3D space. a** Tactile and touchless signals of the FBSS during two-dimensional teaching process, and the time instants of a user teaching the soft manipulator to grasp an object in a touchless manner. **i** Before bending & grasping the soft manipulator, the user chose an adequate length of the soft manipulator by controlling the distance between the hand and the FBSS (orange shading). **ii** The user initially applied a large step size by putting a hand very close to the FBSS (blue shading), then in step **iii** applied a small step control by keeping a short distance from the FBSS (green shading). **iv** When the soft gripper reaches the object, the user pressed the FBSS "button" to grasp the object (purple shading). **b**–**d** Comparison of teaching and repeating trajectories while grasping objects in three different positions. **e** Tactile and touchless signals of FBSS I and II during three-dimensional teaching, and time instants of a user touchlessly teaching the soft manipulator to grasp an object in three-dimensional space. **i** The user controlling planar bending in large steps with left-hand approaches toward the FBSS I (blue shading). **ii** The user controlling planar bending in small steps (also with left-hand approaches toward FBSS I) (purple shading). **iii** The user controlling out-of-plane bending with right-hand approaches toward FBSS II (green shading). **iv** When the soft gripper reaches the object, the user presses the FBSS "button" to grasp the object (orange shading). **f**–**h** Comparison of teaching and repeating trajectories while grasping objects in three different positions. The trajectories are illustrated in three-dimensional space (x–y, y–z, and x–z planes).

participants' positioning errors were less than 10 mm. This result suggests that non-specialists can quickly learn how to position the soft manipulator accurately through interactive teaching.

To enable the interactive teaching of the soft manipulator with even more complex locomotion, we proposed the "shifting sensors and teaching" method (Fig. 5a and Supplementary Fig. 13). Specifically, the FBSS was placed on a flexible, arc-shaped patch with three magnets behind it. Several small magnetic cylinders were placed around the bottom of each segment of the soft manipulator. With the magnetic attachment, the FBSS can be shifted to different positions on the soft manipulator in a rapid, accurate manner. Therefore, the human demonstrator can select a segment for interaction, easily shift the FBSS patch to the corresponding segment and then teach the soft manipulator in a touchless manner. Thus, we name this method "shifting sensors and teaching".

With the proposed "shifting sensors and teaching" method, we show the interactive teaching of the soft manipulator with complex locomotion in 2D and 3D spaces. The normalized touchless and tactile signals of FBSS I and FBSS II are also plotted against time (Fig. 7).

With the "shifting sensors and teaching" method, a user interactively taught the soft manipulator to achieve a 2D "S" shape (Fig. 7a and Supplementary Movie 16). In step (i), two FBSSs were placed on the bottom of the third segment of the soft manipulator. When the demonstrator's two hands approached the two FBSSs simultaneously, all three segments of the soft manipulator shortened and entered the teaching mode. In step (ii), the demonstrator shifted the FBSS I to the right side of the first segment and then used their right hand to bend the first segment to the left. Then the demonstrator pressed the FBSS I to lock the first segment (iii). In step (iv), the demonstrator shifted the FBSS II to the second segment's left side, used their left hand to bend the second segment to the right, and then pressed FBSS II to lock the second segment. In step (v), the FBSS I was shifted to the right side of the third segment. The demonstrator used the right hand to bend the third segment to the left then pressed FBSS I to lock the third segment and finished the touchless teaching session. According to this method, we realized a planar "S"-shaped configuration of the soft manipulator using the "shifting sensors and teaching" method by shifting the FBSS sensors three times.

We show an interactive teaching session involving complex locomotion in 3D space by applying the "shifting sensors and teaching" method (Fig. 7b and Supplementary Movies 17, 18). In step (i), the soft manipulator was triggered to enter the teaching mode. In step (ii), the demonstrator shifted the FBSS II to the right side of the first segment and then used the right hand to bend the first segment to the left. Then the demonstrator pressed the FBSS II sensor to "lock" the first segment in the current direction (iii). In step (iv), FBSS I was shifted to the back of the first segment, and the right hand "bent" the soft manipulator to move outward. Then the first segment was "locked" by pressing the FBSS I. In step (v), the FBSS II was shifted to the left side of the second segment. The demonstrator used the left hand to bend the second segment to the right and "locked" the second segment in the current direction by pressing the FBSS II. In step (vi), the FBSS I was shifted to the front of the second segment. Then

the demonstrator used the right hand to bend the second segment inward and "locked" the second segment by pressing the FBSS I. In the final step (vii), the FBSS II was shifted to the left side of the third segment. The demonstrator used the left hand to bend the third segment toward the right then pressed FBSS II to lock the third segment, and finished the touchless teaching session. Thus, we realized a complex 3D configuration (note that all nine chambers of the soft manipulator were involved) of the soft manipulator using the "shifting sensors and teaching" method by shifting the FBSS sensors five times. These teaching processes took 197 and 350 s, respectively. The experimental results show that the "shifting sensors and teaching" method is simple and effective in enabling complex 3D configurations of soft continuum robots.

We show that a human can interact closely with the soft manipulator to complete another challenging task. The watercolor pen was installed at the end of the manipulator (Fig. 8a). With this setup, we taught the manipulator to execute movements to "navigate" a maze on paper (Fig. 8b and Supplementary Movie 19). The soft manipulator repeated the trace after teaching (Fig. 8c). The output signals of FBSS I and II were recorded over the ~240 s teaching period (Fig. 8d).

We also show the manipulator's ability to perform a critical task in the context of public health. As the coronavirus pandemic continues to range around the world, throat swabs have become a common practice for medical testing. However, this undoubtedly has put a burden on medical workers who risk infection during the collection process. To address this issue, we used the interactive system to teach the soft manipulator to take a throat swab (Fig. 8e, f and Supplementary Movie 20). First, a cotton swab was installed at the end of the soft manipulator. The user could then touchlessly bend the first two segments of the soft manipulator by hand to control the position of the swab. Once the swab reached the target position, the user elongates the soft manipulator's third segment by pressing the FBSS to and collect the throat swab sample. The intelligent interactive system is simple enough for medical workers to use without extensive training, and the soft manipulator can repeat the action autonomously after only a single teaching process. Compared with traditional rigid robots, soft manipulators are inherently safer for human interaction, because of their soft materials and compliant structures. The real-time driven air pressure was compared during the teaching and repeating processes (Supplementary Fig. 14). The two air pressure curves show almost identical changes during both processes.

Finally, we show that the soft manipulator can be "taught" to cross a barrier and successfully grasp an artificial flower by shifting the FBSSs five times (Fig. 8g and Supplementary Movies 21, 22). To cross the barrier, we touchless controlled the third segment to bend outward (i) and the first segment to shorten. Then the second segment was bent to the right (ii), and the third segment was bent upward (iii) and inward. To grasp the flower, the third segment was bent downward (iv) and the gripper grasped the flower by pressing the FBSS II (v), and the whole process lasted about 318 s. The experimental result shows the advantages of the "shifting sensors and teaching" method in the practical application of soft robot multi-degree-freedom control, and provides a new scheme for multi-degree-freedom control of the soft robot.

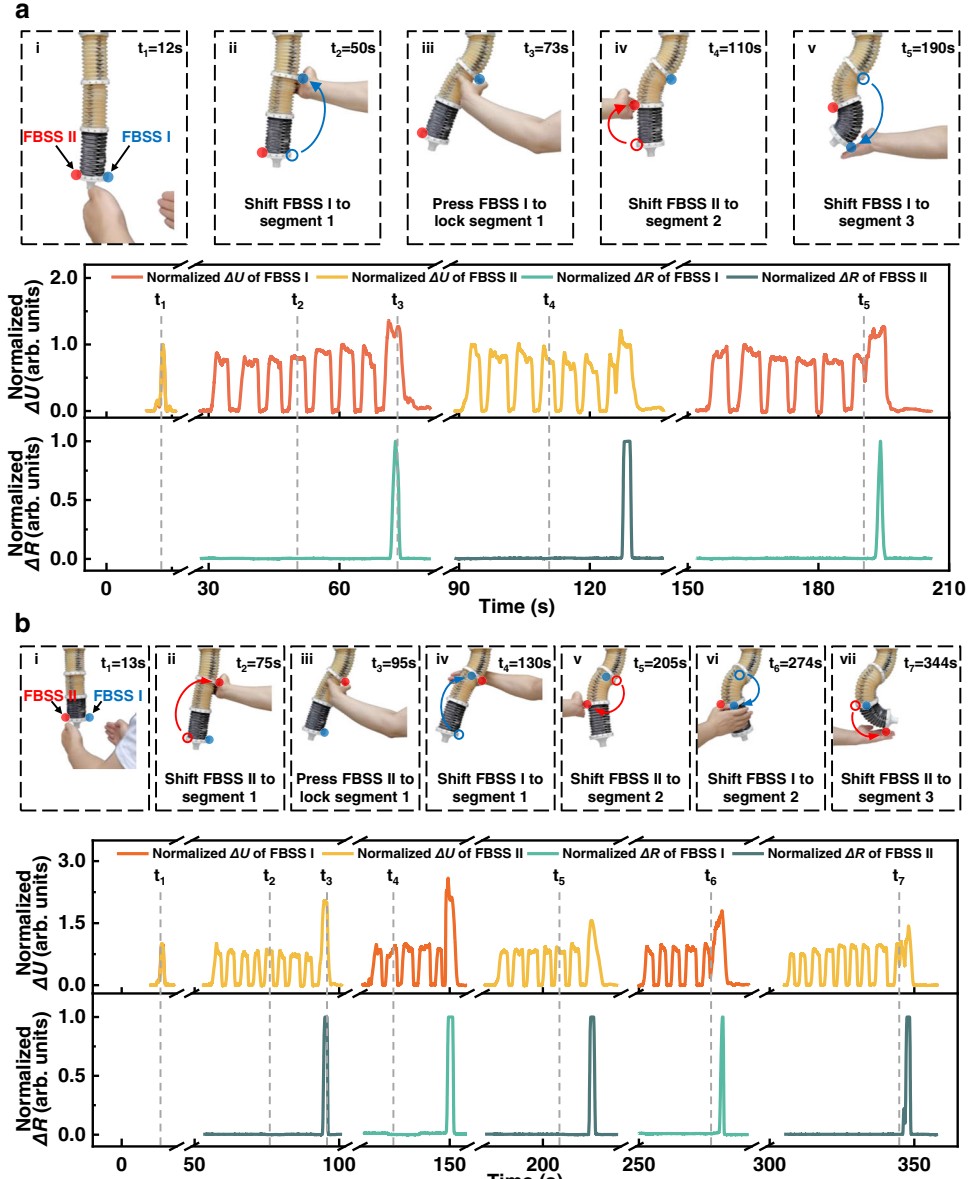

**Fig. 7 | Interactive teaching soft manipulator performs complex locomotion based on the "shifting sensors and teaching" method.** The normalized touchless and tactile signals of FBSS I and FBSS II versus time. The red and blue arrows in each panel indicate that the user was moving the FBSS from one position to another in order to interact differently with the soft manipulator. **a** The demonstrator teaches two-dimensional movements using the "shifting sensors and teaching" method. **b** Interactively teaching complex three-dimensional locomotion by applying the "shifting sensors and teaching" method.

## Discussion

In this paper, we developed a flexible bimodal smart skin (FBSS) prototype that responds to tactile and touchless stimulations and distinguishes between the two modes in real time. With the FBSS as an interface, we proposed a human-soft robot touchless interactive teaching method and systematically tested this method on a continuum soft manipulator. This interactive teaching method for executing complex motions was realized intuitively via bare-handed touchless approaches to the FBSS. Using this method, we successfully taught a "naive" soft manipulator to move in 3D space and perform simple tasks such as painting, taking a throat swab, and crossing a barrier to grasp an object. We envision that this interactive teaching method may expand the practical uses of soft robots, as it allows non-specialists to operate the soft robot for various tasks without expert familiarity.

In terms of other relevant sensing methods, there are a few other sensors that can detect tactile and touchless stimulations. For

example, magnetic bimodal skin relies on the giant magnetoresistance (GMR) effect via a magnetic film with a pyramid-shaped extrusion at its top surface[50]. When the giant magnetoresistance sensor detects a magnetic field around the film, its resistance changes. However, this sensor requires objects contacting the film to be magnetic, so the material properties of the sensed objects are quite limited. In contrast, a triboelectric sensor can detect a wide range of materials.

The high flexibility and sensitivity of triboelectric electronic skin enabled the soft robot to have both sensing and interaction capabilities. Flexible triboelectric skin, which mainly includes flexible electric and dielectric films[51,52], can convert tactile and touchless stimulations into voltage signals through contact electrification and electrostatic induction, respectively. It should be noted that the tactile and touchless stimulations result in the identical trend of electric variation, it is hard for triboelectric skins to distinguish those two modes in real-time[43,44]. By combining triboelectric and liquid metal sensory mechanisms, our FBSS can simultaneously sense both tactile

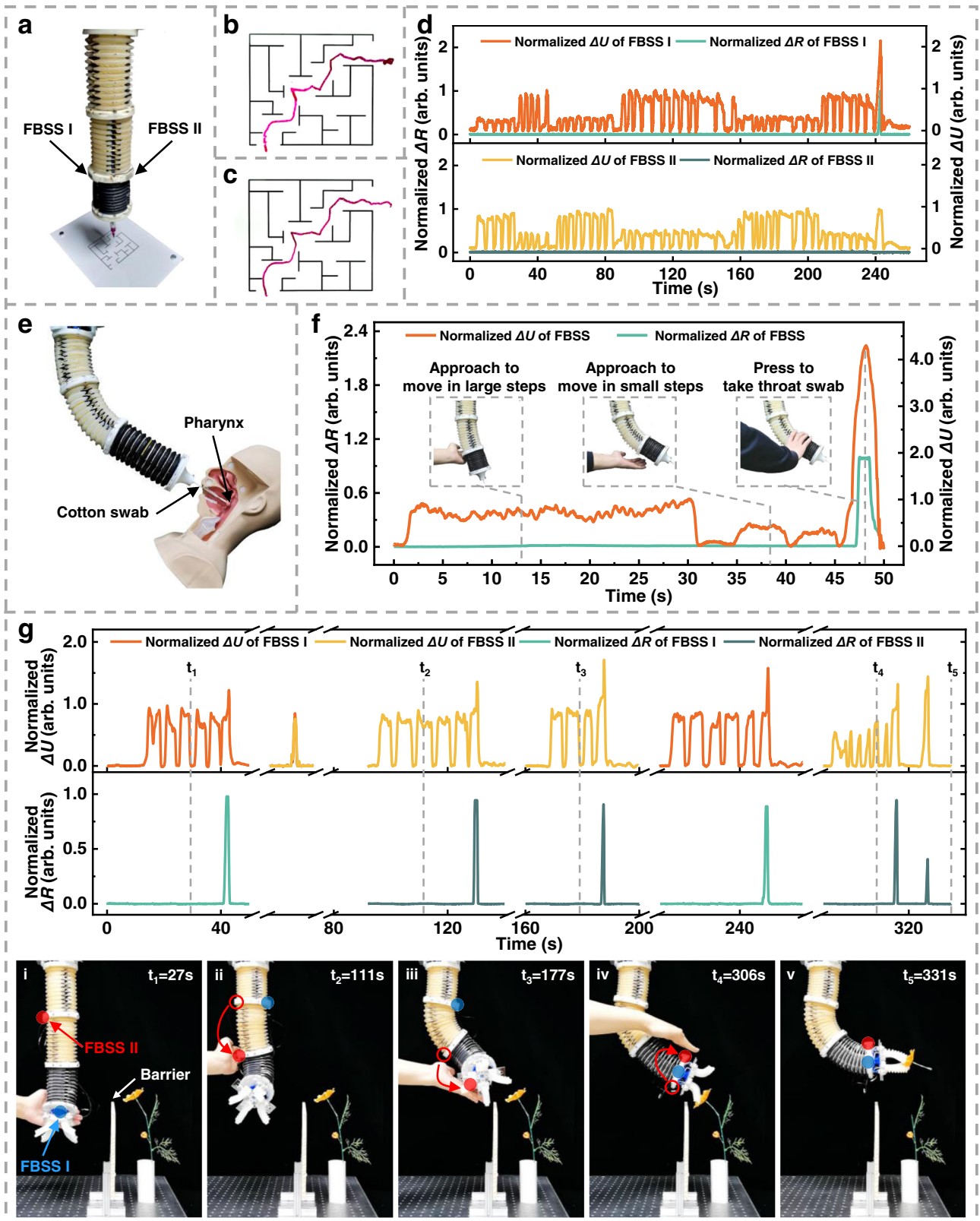

**Fig. 8 | Demonstration of potential applications for interactively teaching self-reacting soft robots.** **a** Photograph of the experimental setup for teaching a soft manipulator to complete a maze. **b**, **c** The manipulator's maze traces during teaching and repeating processes. **d** Signal curves during the maze completion teaching process. **e** Photograph and **f** signal curves of interactive teaching the soft manipulator to take throat swab. **g** Interactively teaching the soft manipulator to cross a barrier and successfully grasp an artificial flower.

and touchless information and can distinguish the two modes in real time. Since the FBSS is flexible and stretchable, it is suitable for large deformation and can be used for soft robotic sensing. In addition, the FBSS can sense a wide range of materials during interaction (Fig. 3c). We compared the FBSS with other tactile/touchless sensors (Supplementary Table 4). In the future, the FBSS sensing accuracy can be further improved by optimizing the microscopic pyramid structure and the liquid metal channels.

There are few previous reports on the human interactive teaching of soft robots. Our proposed methodology has the following unique features for teaching movement: (1) the method is based on a touchless, near-distance control via the natural hand-eye coordination of the human participants, making it intuitive and straightforward; and (2) the teaching result is quite effective in terms of time and accuracy. With the touchless interactive teaching method, most 3D movements of the soft manipulator were finished within a few minutes.

The "contact" teaching approach for traditional rigid robotic manipulators (i.e., achieving compliant behavior with a robot's end effector in response to forces exerted by a human operator) is problematic for teaching soft robots[46]. Capturing the passive deformation of soft continuum manipulators, for example, when manipulated by a human hand requires many soft sensors, which complicates shape and configuration reconstruction later on. In contrast, human subjects without any robotic teaching skills or experience have validated the practicality and effectiveness of this teaching method. In addition, recording and analyzing the touchless/tactile information during the teaching process and replaying the soft robot's movements can help illustrate how humans prefer to touchless interact with robots. A record of the hand movements, particularly when approaching the robot, of different individuals would further help to refine the interactive system.

In terms of the limitations of this research, we used two soft sensors to create an interactive interface in the present study. In future work, we will incorporate more interactive soft sensors into the soft manipulator to enable more complex controls of the robot's shape and to capture more feedback. Furthermore, developing multi-sensor arrays with FBSS and incorporating emerging machine-learning tools would enable more complex robotic motions with different morphologies established through touchless interaction. For example, collect massive sensory data for ML training to recognize human gestures and environmental objects for soft robots. In addition, the ability to distinguish between a human hand and an object or obstacle with FBSS in a complex environment would further complement the current teaching method.

In this study, we used a pneumatically actuated soft robot, which is simple, repeatable, and robust. Responsive materials can allow soft robots to actuate through a variety of stimuli, such as light, magnetic fields, electricity, and chemicals[53], and soft material structures such as origami and metamaterials can enable complex movement through a touchless interactive teaching method[54]. We envision that interactive soft robots can work collaboratively with an increasing number of human participants in a wide variety of disciplines.

## Methods
### Fabrication process of the flexible electrode layer
A patterned Kapton film was first affixed to a clean silicon wafer as a shadow mask (Supplementary Fig. 15A). The Ag NW network solution was then sprayed onto the wafer and the solvent evaporated at 60 °C for 15 min. A thin silicone rubber layer was spin-casted onto the wafer and cured at 60 °C for 4 h. The cured silicone rubber was then carefully peeled off from the wafer and the Ag NW network was transferred onto the silicone rubber.

### Fabrication process of the flexible dielectric layer
The SLA mold with micro-pyramid caves (depth = 320 μm; width = 500 μm) was first printed by a micro-precision 3D printer

(Supplementary Fig. 15B). The silicone rubber (Smooth-on, Dragon skin 00–20) was then drop-casted onto the mold and cured at room temperature for 4 h. The cured silicone rubber was carefully peeled off from the mold and the micro-pyramid structures were transferred onto it.

### Fabrication process of the liquid metal patch
Using a liquid metal printer (DREAM Ink, DP-1), the patterned liquid metal was printed on the plastic substrate (Supplementary Fig. 16). The silicone rubber (Smooth-on, Dragon skin 00-20) was drop-casted onto the patterned liquid metal and then cured at room temperature for 4 h. To transfer the patterned liquid metal from the substrate onto the silicone rubber, they were placed in a refrigerator at −140 °C for 40 min. The cured silicone was carefully peeled off from the substrate and the patterned liquid metal was embedded in the silicone. Additional silicone rubber (Smooth-on, Dragon skin 00-20) was drop-casted onto another side of the patterned liquid metal and cured at room temperature for 4 h.

### Implementation and control of the interactive soft manipulator
The soft manipulator is designed and fabricated for grasping objects (Supplementary Fig. 17). The soft manipulator primarily consists of three soft actuator modules and a soft gripper as the end effector (Fig. 5a). The soft manipulator has 10 pneumatic chambers and each bending segment has 3 chambers (with 3 bending segments in total). The end effector, a four-fingered soft gripper, is actuated by a single air inlet.

During the teaching process, the motion of the soft manipulator follows a kinematic model under the hypothesis of piecewise constant curvature (PCC). We consider the soft manipulator's shape to be composed of a fixed number of segments with constant curvature. The kinematic model transforms from configuration space (arc parameters $\kappa_i$, $\theta_i$, $\varphi_i$) to actuation space (chamber pressure $p_{ij}$) More specifically, the indices $i = 1, 2, 3$ and $j = 1, 2, 3$ refer to the $i$th segment and the $j$th chamber, respectively. We define the arc parameter $\kappa_i$ as the curvature radius of the $i$th segment, $\theta_i$ as the curvature angle around the y-axis, and $\varphi_i$ as the deflection angle around the z-axis (Fig. 5b, c). The constant parameter $d$ could be measured before initiating actuation. First, we solve the chamber length $l_{ij}$ from given arc parameters $\kappa_i$, $\theta_i$, $\varphi_i$ ($r_i = \kappa_i^{-1}$), shown in Eq. (1):

$$\begin{cases} l_{i1} = \theta_i \cdot (r_i - d \sin \varphi_i) \\ l_{i2} = \theta_i \cdot \left[r_i + d \cos(\varphi_i - \frac{\pi}{6})\right] \\ l_{i3} = \theta_i \cdot \left[r_i - d \cos(\varphi_i + \frac{\pi}{6})\right] \end{cases} \quad (1)$$

Then adding in calibrated pressure-length relations (Supplementary Fig. 18), we can calculate the actuating pressure $p_{ij}$ from the chamber length $l_{ij}$ to complete the model-based control.

### Interactive control of soft manipulator based on the FBSS interface
The teaching process starts with a human hand approaching the FBSS, and the FBSS transforms the distance information into a voltage signal (Fig. 5d and Supplementary Table 2). First, we normalize the voltage signal through Eq. (2), with a sampling frequency of 100 Hz:

$$S_{out} = \frac{V - V_{init}}{V_{max} - V_{init}} \quad (2)$$

where $S_{out}$ is the normalized voltage signal, $V$ is the measured voltage of the FBSS, $V_{max}$ is the maximum voltage output of the FBSS, and $V_{init}$ is the initial voltage output of the FBSS. Then, we use a 10-time sampling frequency mean filter to remove noise from the signal. To obtain a variable step length from the FBSS feedback signal, we

implement the hyperbolic tangent function as the mapping of the step length (Fig. 5e), shown in Eq. (3):

$$\theta_{\mathrm{h}} = h_{\mathrm{init}} \cdot \tanh\left(k_1^{(1-\frac{1}{k_2}\frac{1}{S_{\mathrm{out}}})}\right) \qquad (3)$$

where $\theta_{\mathrm{h}}$ is the calculated step length, $h_{\mathrm{init}}$ is the initial step length, and $k_1$, $k_2$ are the parameters of the hyperbolic tangent function. Then, the step length is added to the current bending angle $\theta_0$ in Eq. (4):

$$\theta = \theta_0 + \theta_{\mathrm{h}} \qquad (4)$$

The bending angle $\theta$ is substituted into the kinematic model to solve the chamber pressures $p_{ij}$ of the soft manipulator, and the multi-channel pneumatic system executes actuation with pressures $p_{ij}$. Finally, when the FBSS registers physical contact, the soft manipulator triggers the gripper to complete the gripping motion.

During the contact-free interaction process, the end effector of the soft manipulator can be guided by the motion of the human hand in 3D space. The position of the end effector can be determined by the operator's vision.

We evaluated the teaching error under different step length control strategies. We mounted a laser transmitter on the end effector of the soft manipulator to track the position of the end effector. During the entire interaction process, time cost and position error are recorded, and each test is repeated 5 times. The average execution time and position error of the variable step length strategy are 24.3 s and 4.6 mm, respectively (Fig. 5f). Thus, we adopted the variable step length strategy for interactive teaching. The workspace of the soft manipulator is 568 mm in length, 591 mm in width, and 334 mm in height (Fig. 5g). This guarantees a wide range of interactive locomotion for a human user.

### Electrical characterization

The resistance of the FBSS was measured with a synchronous data acquisition card (National Instruments, USB-6356). The voltage, current, and transfer charges were measured using an electrometer (Tektronix Inc., Keithley 6514).

## Data availability

The data generated in this study are provided in the Source data file. Source data are provided with this paper.

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

## Acknowledgements

This work was supported by the National Science Foundation support projects, China (Grant Nos. 91848206, 92048302, T2121003 received by L.W.), and the National Key R&D Program of China (Grant Nos. 2018YFB1304600, 2019YFB1309600, 2020YFB1313003 received by L.W.). We want to thank Zhexin Xie, Shiqiang Wang, Shanshan Du, and Chuqian Wang for their assistance in this work.

## Author contributions

W.L. and L.W. conceived the idea, W.L., Y.D., J.L., F.Y., C.Z., Y.W, B.F., F.S., X.D., and L.W. analyzed the data and wrote the paper. W.L. designed and fabricated the FBSS. Y.D. and J.L. designed and fabricated the soft manipulator. W.L., Y.D., J.L., and H.Y. implemented the interactive teaching system. Lei L., G.W., B.C., S.W., Luchen L., H.Y., and Y.L. conducted the experiments. Lei L., Y.M., and W.L. draw and optimized the figures, tables, and videos.

## Competing interests

The authors declare no competing interests.
