## [Peer Review File · Nature Communications]

REVIEWER COMMENTS

Reviewer #1 (Remarks to the Author):

This is an excellent work about the interactive teaching of soft robots through flexible touchless and tactile bimodal sensory interfaces. The authors described a flexible bimodal smart skin with a triboelectric nanogenerator and liquid metal sensors. The proposed FBSS can distinguish the touchless and tactile modes in real-time. With the FBSS, the authors also investigated an interactive teaching method in both tactile and touchless manners. This method allows a non-specialist to teach a soft continuum robot multiple tasks efficiently and interactively.

The results are promising. I believe this manuscript will interest a wide range of readers. The manuscript is well written. I think the manuscript can be accepted after addressing the following minor concerns:

1. To better clarify the working principle of the FBSS, how the production of the surface charges on the external object should be more clearly described.
2. The data points in Fig. 3a, b, and c are not very clear; their sizes should be amplified. Also, the position of the FBSS should be marked out in Fig. 3g.
3. The inset of Fig. 5d and its background is not clear, and the position of the FBSS should be pointed out.
4. Please amplify the inset of Fig. 6d or introduce more details about its experimental process in the Supplementary materials.
5. According to Fig. 7a, the human can realize independently interactive control of the soft manipulator in both directions. How to avoid the crosstalk of the touchless signals of the two FBSS equipped on the soft manipulator?
6. The human teaching soft manipulator taking a throat swab section is too brief; it would be better to elaborate more about this section.

Reviewer #2 (Remarks to the Author):

This paper presents a very interesting topic on how to teach soft robots to act. For rigid robotic arms, drag and drop is widely adopted to quickly program them for non-experts, yet for soft robots, this strategy is not possible due to infinite configurations of their soft and continuum body. This paper proposed to use tactile and touchless sensors to detect human's intention and certain motions, and then "teach" the soft robotic arm to move in corresponding manners. Overall, I believe this is a very smart and inspiring idea to finally put soft robots into practical applications. In addition, this paper includes a thorough characterization of components and nice demonstrations. I highly recommend this paper to be considered in Nature Communications. To further improve the quality of this paper, I have the following concern for the authors to address.

1. One important aspect of the paper is the multimodal sensor. Currently, the sensors have been characterized carefully, however, some important aspects that are related to their practical usage are still missing, including their resolution, signal-to-noise ratio, and dynamic response (or bandwidth). If the sensor has insufficient bandwidth, when used in the teaching scenario, the participant might feel a lag in its responses. More specifically, Fig. 3a-c needs error bars to show how repeatable the sensor is.
2. The touchless sensor has shown very nice performance, however, I'm a bit worried about how it reacts to environmental change, such as electromagnetic interferences, humidity, temperature, etc. Please supplement such data or literature.
3. I have seen that the robot can successfully accomplish several rather complex tasks through the proposed teaching method. The robot has a total of 9 chambers excluding the gripper, currently, the 3 DoFs have been utilized to achieve a rather complex 3D manipulation. Please comment on how the 9 DoFs all be utilized in the teaching algorithm to further show the full advantage of the soft, continuum robot.

Reviewer #3 (Remarks to the Author):

In this work, the authors presented a multimodal flexible sensory interface for interactively teaching soft robots to perform skilled locomotion using bare human hands. Especially, the author developed a flexible bimodal smart skin (FBSS) based on triboelectric nanogenerators and liquid metal sensors. Using triboelectric nanogenerators for tactile sensing is a very old story, which could be dated back to the year 2013. And there are many other reports in the past 9 years, including but not limited to,

1. Nano Energy 2019, <https://doi.org/10.1016/j.nanoen.2019.02.054>
 2. Advanced Functional Materials, 2018 <https://doi.org/10.1002/adfm.201802989>
 3. Advanced Functional Materials, 2019 <https://doi.org/10.1002/aelm.201901174>
 4. ACS Nano 2017 <https://pubs.acs.org/doi/abs/10.1021/acsnano.7b00396>
 5. Materials Today Energy 2021, <https://doi.org/10.1016/j.mtener.2021.100657>
 6. Nano Energy 2021, <https://www.sciencedirect.com/science/article/pii/S2211285520311630>
- , and many others.

What is more, there are many reports on using soft triboelectric nanogenerators tactile sensing for robotics, which is exactly the same as the current report. For example, just naming a few as follows:

1. Advanced Functional Materials, 2019 <https://doi.org/10.1002/adfm.201907312>
2. Nano Energy 2019, <https://doi.org/10.1016/j.nanoen.2019.104005>
3. Nature Communications 2020, <https://www.nature.com/articles/s41467-020-19059-3>

I have to say that there is not much technical or fundamental advancement in this manuscript that deserves a publication in Nature Communications. Additionally, the manuscript is poorly written and organized. It is suggested to well revise the manuscript before any future submissions.

Responses to the reviewers' comments

Manuscript ID: NCOMMS-22-10862

Manuscript Type: Article

Title: Interactive teaching of soft robots through flexible touchless and tactile bimodal sensory interfaces

Author(s): Wenbo Liu^{1,†}, Youning Duo^{1,†}, Jiaqi Liu^{1,†}, Feiyang Yuan^{1,†}, Lei Li¹, Luchen Li¹, Gang Wang¹, Bohan Chen¹, Siqi Wang¹, Hui Yang², Yuchen Liu³, Yanru Mo³, Yun Wang¹, Bin Fang⁴, Fuchun Sun⁴, Xilun Ding¹, Chi Zhang^{5,6}, Li Wen^{1*}

Corresponding author: Li Wen, liwen@buaa.edu.cn

[†]These authors contributed equally to this work.

We appreciate your insightful comments as they help notably improve our paper's quality. We have made our maximum efforts to improve the manuscript and address your comments. Significant changes have been made in the revised manuscript. Here we provided the detailed response letter to the comments from the reviewers.

Reviewer #1:

This is an excellent work about the interactive teaching of soft robots through flexible touchless and tactile bimodal sensory interfaces. The authors described a flexible bimodal smart skin with a triboelectric nanogenerator and liquid metal sensors. The proposed FBSS can distinguish the touchless and tactile modes in real-time. With the FBSS, the authors also investigated an interactive teaching method in both tactile and touchless manners. This method allows a non-specialist to teach a soft continuum robot multiple tasks efficiently and interactively.

The results are promising. I believe this manuscript will interest a wide range of readers. The manuscript is well written. I think the manuscript can be accepted after addressing the following minor concerns:

Response: Thank you very much for your positive comments. We will make persistent efforts.

1. To better clarify the working principle of the FBSS, how the production of the surface charges on the external object should be more clearly described.

Response: Many thanks for the question! We have added more details of the production process of the surface electric charges on the external object and FBSS

(Supplementary Figure 2). We have also modified the **Results** section “*Working principle and sensing performance of FBSS*” below.

“During the initial stage (i), equal negative and positive charges are generated on the flexible dielectric layer and external object from different electron affinities after a few repeated contacts. These surface charges can remain for a sufficient time (over 1 hour) for the interactive teaching process (Supplementary Fig. 2).”

Supplementary Figure 2: Surface charge production on the external object and FBSS.

In the initial state (i), there is no electric charge on the surface of the external object (red) and the flexible dielectric layer (cyan). In the second state (ii), equal negative and positive charges were generated on the flexible dielectric layer (cyan) and external object (red) from different electron affinities after a few repeated contacts. In the third state (iii), the external object (red) was separated by the flexible dielectric layer (cyan), and these surface charges can remain a sufficient time for the interactive teaching process.

2. The data points in Fig. 3a, b and c are not very clear; their sizes should be

amplified. Also, the position of the FBSS should be marked out in Fig. 3g.

Response: Thanks for the valuable suggestion! We have amplified the data points in Fig. 3a, b, and c. Related changes have been made in the revised Fig. 3.

Fig. 3. Characterization of the FBSS for tactile and touchless sensing. a Tactile

(cyan) and touchless (orange) output signals tested under different distances between a surface (glass) and the FBSS sensor. **b** Tactile and touchless output signals under different loading pressure. The supplementary materials and methods provided more details about the loading experiments. **c** FBSS's output signals at surfaces with different materials (with an above distance of 20 mm). All error bars represent ± 1 SD, and $N=5$.

In addition, the position of the FBSS have been marked out in Fig. 3j according to your suggestion.

Fig. 3. Characterization of the FBSS for tactile and touchless sensing. j, k Image

and tactile and touchless output signals of the FBSS when a human finger press on it.

Through a program, a red light-emitting diode (LED) was turned on when the induced touchless signal exceeded a threshold value; the blue LED was turned on when the finger touched the FBSS.

3. The inset of Fig. 5d and its background is not clear, and the position of the FBSS should be pointed out.

Response: Many thanks for your constructive suggestion! We have replaced the background of Fig. 5d, and position of the FBSS has been pointed out. The revised Fig. 5 and its caption are as follows:

Fig. 5. Interactive teaching kinematics control method of the soft manipulator. d

The closed-loop control framework for interactive teaching of the soft manipulator, where V represents the measured voltage of the FBSS, S_{out} is the normalized voltage signal, θ_h is the calculated step length, and p_{ij} represents the pneumatic pressure of each soft actuator.

4. Please amplify the inset of Fig. 6d or introduce more details about its experimental process in the Supplementary materials.

Response: Thanks for the valuable suggestion! We have put the Fig. 6d in the Supplementary Information:

Supplementary Figure 12: Teaching experiment with multiple participants.

(A) The principle of measuring the position error in experiments. A laser pointer is attached to the soft manipulator and participants can control the position of the laser point by touchless teaching the soft manipulator. The position error is defined as the distance between the final position of the laser and the center of the target. (B) The positioning error of the manipulator after being taught by multiple human subjects, including two experts and three novices.

5. According to Fig. 7a, the human can realize independently interactive control of the soft manipulator in both directions. How to avoid the crosstalk of the touchless signals of the two FBSS equipped on the soft manipulator?

Response: Thank you for your question! We utilized two approaches to avoid the

crosstalk of the two FBSS sensors. Firstly, we put the two FBSSs in two perpendicular planes for the interactive teaching, and the two FBSS prototypes have a certain distance (>100 mm). Secondly, in the interactive teaching control method, we use the hyperbolic tangent function to map the normalized voltage signal S_{out} to the step length θ_h , as shown in Fig. 5(e). We can change the parameters k_1 and k_2 to adjust the shape of the S_{out} . The step length mapped from the small interference signals is almost zero by making the curve's initial stage flat. Thus, the soft manipulator would not interfere with the signals from human hands. These two approaches can avoid the crosstalk between the two FBSSs and ensure the independently interactive control of the soft manipulator.

6. The human teaching soft manipulator taking a throat swab section is too brief; it would be better to elaborate more about this section.

Response: Thank you for your constructive suggestion! We have added a more detailed description of the throat swab experiment in the Results section “*Interactive teaching of the soft manipulator*”, shown as below:

“First, a cotton swab was installed at the end of the soft manipulator. The user could then touchlessly bend the first two segments of the soft manipulator by hand to control the position of the swab. Once the swab reached the target position, the user elongates the soft manipulator's third segment by pressing the FBSS to and collect the throat swab sample.”

Reviewer #2:

This paper presents a very interesting topic on how to teach soft robots to act. For rigid robotic arms, drag and drop is widely adopted to quickly program them for non-experts, yet for soft robots, this strategy is not possible due to infinite configurations of their soft and continuum body. This paper proposed to use tactile and touchless sensors to detect human's intention and certain motions, and then "teach" the soft robotic arm to move in corresponding manners. Overall, I believe this is a very smart and inspiring idea to finally put soft robots into practical applications. In addition, this paper includes a thorough characterization of components and nice demonstrations. I highly recommend this paper to be considered in Nature Communications. To further improve the quality of this paper, I have the following concern for the authors to address.

Response: Thank you very much for your positive comments on the manuscript! All responses to your valuable suggestions are listed below.

1. One important aspect of the paper is the multimodal sensor. Currently, the sensors have been characterized carefully, however, some important aspects that are related to their practical usage are still missing, including their resolution, signal-to-noise ratio, and dynamic response (or bandwidth). If the sensor has insufficient bandwidth, when used in the teaching scenario, the participant might feel a lag in its responses.

More specifically, Fig. 3a-c needs error bars to show how repeatable the sensor is.

Response: Thanks very much for the valuable suggestion! We have accordingly added the experiments of the resolution, signal-to-noise ratio, and dynamic response for the FBSS. In the revised manuscript Results section “Working principle and sensing performance of FBSS”, the third paragraph added the description accordingly:

"The dynamic response of the tactile sensing of the FBSS is about 120 ms, which is close to that of human skin (Supplementary Fig. 4A, B). The tactile and touchless signal noises are 0.04 Ω and 0.12 V, respectively (Supplementary Fig. 5A, B). The maximum Signal-to-noise ratio (SNR) of the touchless signal is 94.58, and the highest SNR of the tactile signal is 431.03 (Supplementary Fig. 5C, D). The maximum resolutions measured in the touchless and tactile experiments are 0.05 mm and 0.35 kPa, respectively (Supplementary Fig. 6A, B)."

We have added the experimental results in the Supplementary Fig. 4-6.

Supplementary Figure 4. The dynamic response of the FBSS.

(A) The experimentally measured change of the resistance of the FBSS under a fast pressure stimulus. (B) A close-up of the area indicated within the dashed box in (A).

Supplementary Figure 5. The signal-to-noise ratio (SNR) of the FBSS.

(A), (B) The noise of tactile and touchless signals. (C), (D) The SNRs of the touchless and tactile signals.

Supplementary Figure 6. The resolution of the FBSS. (A) The resolution of touchless sensing. The distance between the testing surface (glass) and the FBSS gradually decreases from 0.5 to 0 mm. **(B)** The resolution of tactile sensing. The indenting pressure imposed on the sensor gradually increase from 2.00 to 5.52 N.

According to your suggestion, we have added error bars in Fig. 3a-c.

Fig. 3. Characterization of the FBSS for tactile and touchless sensing. a Tactile (cyan)

and touchless (orange) output signals tested under different distances between a surface

(glass) and the FBSS sensor. **b** Tactile and touchless output signals under different loading pressure. The supplementary materials and methods provided more details about the loading experiments. **c** FBSS's output signals at surfaces with different materials (with an above distance of 20 mm). All error bars represent ± 1 SD, and $N=5$.

2. *The touchless sensor has shown very nice performance, however, I'm a bit worried about how it reacts to environmental change, such as electromagnetic interferences, humidity, temperature, etc. Please supplement such data or literature.*

Response: Thank you for your valuable comments! According to your comments, we have experimentally tested the effects of the electromagnetic interferences, humidity, and temperature on the FBSS. We have added the experimental results in the revised **Results** section “*Working principle and sensing performance of FBSS*”, the fourth paragraph:

“To evaluate how environmental factors affect the sensing performance of the FBSS, we experimentally tested the effects of temperature, humidity, and electromagnetic interference on the FBSS. The output touchless signal increases as the temperature increases from 15 to 30 °C, and then remains stable with further temperature increases (Fig. 3d). The output tactile signal remains almost invariant with an increase in temperature. We investigated the effect of humidity on the output signals of the FBSS (Fig. 3e). The touchless signal decreases gradually as humidity increases from 31.4% to 71.4%. The

output tactile signal remains almost invariant with an increase in humidity.

The touchless and tactile signals remain unchanged with an increase in

electromagnetic interference (Fig. 3f). The long-term stability of the FBSS is

also validated under an external pressure of 10 kPa and a distance of 20 mm.

We measured outputs of the FBSS over 1,200 cycles in the same condition

(Fig. 3g). The results show no obvious waveform changes, which points to

the long-term usage of the FBSS.”

Fig. 3. Characterization results of the proposed FBSS prototype for tactile and

touchless sensing. The reactions of the FBSS to environmental change, including **d**

temperature, **e** humidity, and **f** electromagnetic interference.

3. I have seen that the robot can successfully accomplish several rather complex tasks

through the proposed teaching method. The robot has a total of 9 chambers excluding

the gripper, currently, the 3 DoFs have been utilized to achieve a rather complex 3D manipulation. Please comment on how the 9 DoFs all be utilized in the teaching algorithm to further show the full advantage of the soft, continuum robot.

Response: Thanks for the insightful comments! We fully agree with you that more chambers of the soft manipulator should be used during the teaching process. To demonstrate the advantages of the teaching method in the multi-degree-freedom control of the continuum soft robot, we conducted more experiments regarding interactive teaching of the soft manipulator complex locomotion. The experiments include i) rapidly switching the positions of FBSS sensors on the soft manipulator utilizing tiny magnets, thus increasing the interaction points on the soft manipulator. ii) teach the soft manipulator complex three-dimensional locomotion. Up to 9 chambers were used during this process. We also demonstrated that the soft manipulator could be taught to cross the obstacle and grasp an artificial flower. We have added more experimental details in the revised **Methods** section “*Implementation and control of the interactive soft manipulator*”, in the seventh paragraph:

“To enable the interactive teaching of the soft manipulator with even more complex locomotion, we proposed the "shifting sensors and teaching" method (Fig. 5a and Supplementary Fig. 13). Specifically, the FBSS was placed on a flexible, arc-shaped patch with three magnets behind it. Several small magnetic cylinders were placed around the bottom of each segment of

the soft manipulator. With the magnetic attachment, the FBSS can be shifted to different positions on the soft manipulator in a rapid, accurate manner. Therefore, the human demonstrator can select a segment for interaction, easily shift the FBSS patch to the corresponding segment and then teach the soft manipulator in a touchless manner. Thus, we name this method "shifting sensors and teaching".

Fig. 5. Interactive teaching kinematics control method of interactive the soft manipulator. a A schematic view of the soft manipulator, consisting of three segments, each containing three chambers that actuate pneumatically. The FBSS is placed on a flexible, arc-shaped patch with three magnets on the back. Small magnets were also placed around the bottom of each segment of the soft manipulator, so the position of the FBSS can be quickly shifted.

Supplementary Figure 13: Schematic diagram of quickly switching the position of a sensor on a soft manipulator. (A) The photo of the FBSS integrated with the soft manipulator with small magnets. **(B)** The position of the FBSS can be quickly shifted with one hand.

We have also added new experiments in the revised **Results** section “*Interactive teaching of the soft manipulator*”, the 8-10, 13th paragraphs:

“With the proposed "shifting sensors and teaching" method, we show the interactive teaching of the soft manipulator with complex locomotion in 2D and 3D spaces. The normalized touchless and tactile signals of FBSS I and FBSS II are also plotted against time (Fig. 7).

With the "shifting sensors and teaching" method, a user interactively taught the soft manipulator to achieve a 2D “S” shape (Fig. 7a and Supplementary Movie 16). In step (i), two FBSSs were placed on the bottom of the third segment of the soft manipulator. When the demonstrator's two hands approached the two FBSSs simultaneously, all three segments of the soft manipulator shortened and entered the teaching mode. In step (ii), the demonstrator shifted the FBSS I to the right side of the first segment and then

used their right hand to bend the first segment to the left. Then the demonstrator pressed the FBSS I to lock the first segment (iii). In step (iv), the demonstrator shifted the FBSS II to the second segment's left side, used their left hand to bend the second segment to the right, and then pressed FBSS II to lock the second segment. In step (v), the FBSS I was shifted to the right side of the third segment. The demonstrator used the right hand to bend the third segment to the left then pressed FBSS I to lock the third segment and finished the touchless teaching session. According to this method, we realized a planar "S"-shaped configuration of the soft manipulator using the "shifting sensors and teaching" method by shifting the FBSS sensors three times.

We show an interactive teaching session involving complex locomotion in 3D space by applying the "shifting sensors and teaching" method (Fig.7b and Supplementary Movies 17, 18). In step (i), the soft manipulator was triggered to enter the teaching mode. In step (ii), the demonstrator shifted the FBSS II to the right side of the first segment and then used the right hand to bend the first segment to the left. Then the demonstrator pressed the FBSS II sensor to "lock" the first segment in the current direction (iii). In step (iv), FBSS I was shifted to the back of the first segment, and the right hand "bent" the soft manipulator to move outward. Then the first segment was "locked" by pressing the FBSS I. In step (v), the FBSS II was shifted to the left side of the second segment. The demonstrator used the left hand to bend the second segment to the right and "locked" the second segment in the current direction

by pressing the FBSS II. In step (vi), the FBSS I was shifted to the front of the second segment. Then the demonstrator used the right hand to bend the second segment inward and "locked" the second segment by pressing the FBSS I. In the final step (vii), the FBSS II was shifted to the left side of the third segment. The demonstrator used the left hand to bend the third segment toward the right then pressed FBSS II to lock the third segment, and finished the touchless teaching session. Thus, we realized a complex 3D configuration (note that all nine chambers of the soft manipulator were involved) of the soft manipulator using the "shifting sensors and teaching" method by shifting the FBSS sensors five times. These teaching processes took 197 and 350 s, respectively. The experimental results show that the "shifting sensors and teaching" method is simple and effective in enabling complex 3D configurations of soft continuum robots.”

“Finally, we show that the soft manipulator can be "taught" to cross a barrier and successfully grasp an artificial flower by shifting the FBSS sensors five times (Fig. 8g, and Supplementary Movies 21, 22). To cross the barrier, we touchless controlled the third segment to bend outward (i) and the first segment to shorten. Then the second segment was bent to the right (ii), and the third segment was bent upward (iii) and inward. To grasp the flower, the third segment was bent downward (iv) and the gripper grasped the flower by pressing the FBSS II (v), and the whole process lasted about 318 s. The experimental result shows the advantages of the "shifting sensors and

teaching" method in the practical application of soft robot multi-degree-freedom control, and provides a new scheme for multi-degree-freedom control of the soft robot.

Fig. 7. Interactive teaching soft manipulator performs complex locomotion based on the "shifting sensors and teaching" method. The normalized touchless and tactile

signals of FBSS I and FBSS II versus time were plotted. The cyan dashed arrow indicates that the FBSS sensor was moving to the corresponding position for the interaction with the above segment of the soft manipulator. **a** the demonstrator teaches the soft manipulator's two-dimensional movements using the "shifting sensors and teaching" method. **b** interactively teaching the soft manipulator with complex three-dimensional locomotion by applying the "shifting sensors and teaching" method.

Fig. 8g interactively teaching the soft manipulator to cross a barrier and successfully grasp an artificial flower.

Reviewer #3:

In this work, the authors presented a multimodal flexible sensory interface for interactively teaching soft robots to perform skilled locomotion using bare human hands. Especially, the author developed a flexible bimodal smart skin (FBSS) based on triboelectric nanogenerators and liquid metal sensors. Using triboelectric nanogenerators for tactile sensing is a very old story, which could be dated back to the year 2013. And there are many other reports in the past 9 years, including but not limited to,

- 1. Nano Energy 2019, <https://doi.org/10.1016/j.nanoen.2019.02.054>*
 - 2. Advanced Functional Materials, 2018 <https://doi.org/10.1002/adfm.201802989>*
 - 3. Advanced Functional Materials, 2019 <https://doi.org/10.1002/aefm.201901174>*
 - 4. ACS Nano 2017 <https://pubs.acs.org/doi/abs/10.1021/acsnano.7b00396>*
 - 5. Materials Today Energy 2021, <https://doi.org/10.1016/j.mtener.2021.100657>*
 - 6. Nano Energy 2021, <https://www.sciencedirect.com/science/article/pii/S2211285520311630>*
- , and many others.*

What is more, there are many reports on using soft triboelectric nanogenerators tactile sensing for robotics, which is exactly the same as the current report. For example, just naming a few as follows:

- 1. Advanced Functional Materials, 2019 <https://doi.org/10.1002/adfm.201907312>*

2. *Nano Energy* 2019, <https://doi.org/10.1016/j.nanoen.2019.104005>

3. *Nature Communications* 2020,

<https://www.nature.com/articles/s41467-020-19059-3>

I have to say that there is not much technical or fundamental advancement in this manuscript that deserves a publication in Nature Communications. Additionally, the manuscript is poorly written and organized. It is suggested to well revise the manuscript before any future submissions.

Response:

Many thanks for the comments!

First, we want to clarify that this paper's aim is not "*using triboelectric nanogenerators for tactile sensing*". This paper's primary goal is "*human interactively teach soft robots to perform skilled locomotion through multimodal flexible sensory interfaces*". We aim to do this because soft robots, such as soft continuum arms, are challenging to model and program. Non-specialists often face non-negligible obstacles when working with soft robots to achieve specific movements and perform certain tasks. Thus, the blue picture of this study to teaching soft robots to perform complex motions interactively without programming. We envision that interactive teaching may expand the practical uses of soft robots, as it allows non-specialists to operate the robot for various tasks without expert familiarity.

How can we interactively teach soft robots? Few studies demonstrate soft robots' teaching through human interaction. There are two primary challenges to achieving soft robotic teaching through human interaction: the process requires 1) a multimodal, versatile, and robust flexible sensing device for interactions between a soft robot and human demonstrator; and 2) a user-friendly, non-programmable teaching method to transfer a human demonstrator's instructions to the soft robots.

To solve the above two challenges, in this paper, we first developed a flexible bimodal smart skin (FBSS) with both tactile and touchless sensing by integrating a triboelectric sensor with a liquid metal sensor. A triboelectric sensor can respond to touchless stimulation through electrostatic induction, and the liquid metal sensor can respond to tactile stimulation. On this basis, the implemented FBSS can unambiguously distinguish between tactile and touchless modes in real-time.

Thus, the *triboelectric nanogenerator* *is not* our ultimate research objective, but is one component of our implemented sensory interface for the interaction between humans and soft robots.

We appreciate this reviewer for providing six excellent papers on triboelectric nanogenerators for tactile sensing. We have carefully read them thoroughly.

We need to point out that tactile sensors (based on triboelectric nanogenerators and others) are insufficient for interaction (particularly interactive teaching) between humans and soft robots. This is because the tactile sensors can only implement “contact teaching” of robots, which is unsuitable for soft robots. The reasons are two-fold: 1) unlike the rigid manipulator, a soft continuum manipulator's configuration is

challenging to control explicitly for a user because of the manipulator's infinite degrees of freedom and compliant nature. 2) In practice, contact teaching makes the soft continuum robots produce passive deformation, and measuring the deformed configuration of a soft robot requires a large number of soft sensors (either embedded in or on the robot's surface) to reconstruct its three-dimensional kinematics. Given these challenges, we proposed a flexible touchless and tactile bimodal sensory interface to teach soft robots interactively. We have shown in our paper the sensory interface's principal, implementations, and interactive teaching of the soft robotic manipulators.

As this reviewer listed in these six remarkable works, researchers have used triboelectric nanogenerators for tactile sensing. However, it remains to be investigated how to use TENG to realize tactile and touchless bi-mode sensing in real time. As the tactile and touchless stimulation results in the identical trend of electric variation, thus it is challenging for triboelectric sensors to distinguish between tactile and touchless signals in real-time. To emphasize this point, we have newly added Supplementary Fig. 1 and Supplementary Movie 1 in our revised manuscript. Supplementary Fig. 1 shows that the output signal of the triboelectric sensor alone has the same tendency during the process of the human finger from approaching to pressing the FBSS. It is impossible to distinguish the approaching and pressing by the triboelectric signal alone. In contrast, the implemented FBSS in this work can unambiguously distinguish between tactile and touchless modes in real time. To clarify the novelty of this work and the previous literature, we compared FBSS with other tactile/touchless sensors in Supplementary Table 4.

Supplementary Figure 1: FBSS output signals as a finger approaches and presses

it. (A) The output touchless signal increases as the finger approaches the FBSS, while the output tactile signal is negligible. (B) As the finger presses on the FBSS, the output touchless signal increases further and the output tactile signal starts to increase. It is hard to distinguish between the touchless and tactile modes by the output signal of the triboelectric nanogenerator alone. However, the FBSS based on a triboelectric nanogenerator sensor and a liquid metal sensor can transduce both tactile and touchless stimulations simultaneously and distinguish between the two modes in real time.

Meanwhile, we have experimentally tested the effects of the electromagnetic interferences, humidity, and temperature on the FBSS. We have added the experimental results in the revised **Results** section “*Working principle and sensing performance of FBSS*”, the fourth paragraph:

“To evaluate how environmental factors affect the sensing performance of the FBSS, we experimentally tested the effects of temperature, humidity, and electromagnetic interference on the FBSS. The output touchless signal increases as the temperature increases from 15 to 30 °C, and then remains

stable with further temperature increases (Fig. 3d). The output tactile signal remains almost invariant with an increase in temperature. We investigated the effect of humidity on the output signals of the FBSS (Fig. 3e). The touchless signal decreases gradually as humidity increases from 31.4% to 71.4%. The output tactile signal remains almost invariant with an increase in humidity. The touchless and tactile signals remain unchanged with an increase in electromagnetic interference (Fig. 3f).”

Fig. 3. Characterization results of the proposed FBSS prototype for tactile and touchless sensing. The reactions of the FBSS to environmental change, including **d** temperature, **e** humidity, and **f** electromagnetic interference.

The reviewer also pointed out that "*there are many reports on using soft triboelectric nanogenerators tactile sensing for robotics, which is exactly the same as the current report*", and listed a few papers. We thank the reviewers providing three excellent papers, and have properly cited some of them in our revised manuscript. It worth mentioning that these previous studies utilize soft triboelectric nanogenerators as sensors for determine the contact modes of soft robot. However, the contribution of this paper not simply "*using triboelectric nanogenerators tactile sensing for soft robotics*",

but “*interactive teaching of soft robots’ locomotion using soft sensory interfaces*”.

To our knowledge, there are no previous report on human interactive teaching of soft robots’ locomotion using soft sensory interfaces. In particular, in this research, we proposed a user-friendly, non-programmable teaching method to transfer a human demonstrator’s instructions to the soft robots. More specially, we proposed a distance control method that enables human to teach soft robots via bare hand-eye coordination. As a result, human participants can effectively teach a self-reacting soft continuum manipulator complex motions in three-dimensional space within a few minutes.

To fully demonstrate the advantages of the teaching method in the multi-degree-freedom control of the continuum soft robot, we have added additional designs and experiments to improve the interactive teaching method and demonstrate the capacity of interactively teaching complex three-dimensional locomotion of the soft continuum robot. Specifically, we proposed a "shifting sensors and teaching" method, as shown in newly added **Fig. 5a and Supplementary Fig. 13**. Based on the "shifting sensors and teaching" method, we have added **new Fig.7 and Fig. 8g**, and showed that the demonstrator could interactively teach the soft manipulator complex locomotion in three-dimensional spaces and crossing a barrier to grasp a flower. All nine soft manipulator chambers were involved during the newly added teaching process. We also added six new videos, **Supplementary Movies 16-18, 21 and 22**, to show the interactively teaching and replay of the soft manipulator's complex movements using the "shifting sensors and teaching" method. Related text has been added in the supplementary file and the results section “*Interactive teaching of the soft manipulator*”.

Following your comments, we have carefully reviewed the manuscript, removed the typos, and improved its writing.

We sincerely hope that the above responses can reduce your concerns and hope you can find them satisfactory.

REVIEWERS' COMMENTS

Reviewer #1 (Remarks to the Author):

The authors have made significant efforts and revised the manuscript properly according to previous comments. Therefore I recommend to publish this manuscript in the Nature Communications.

Reviewer #2 (Remarks to the Author):

This is a revised version of a previously-submitted work. The authors have thoroughly addressed my concerns by supplementing:

(1) experimental results of the sensor's resolution, signal-to-noise ratio, and dynamic response.

(2) experimental results of the sensor's environmental sensitivity, including humidity, temperature, and electromagnetic interferences, etc.

(3) A more complex task of teaching the robot to cross the obstacle and grasp an artificial flower.

I believe the added data have further improved the quality of the paper and demonstrated the system's robustness and versatility

I support for its publication in Nature Communications.

Reviewer #3 (Remarks to the Author):

After reading the response letter, I still feel not convinced. I have to say that there is not much technical or fundamental advancement in this manuscript that deserves publication in Nature Communications.

Responses to the reviewers' comments

Manuscript ID: NCOMMS-22-10862A

Manuscript Type: Article

Title: Interactive teaching of soft robots through flexible touchless and tactile bimodal sensory interfaces

Author(s): Wenbo Liu^{1,†}, Youning Duo^{1,†}, Jiaqi Liu^{1,†}, Feiyang Yuan^{1,†}, Lei Li¹, Luchen Li¹, Gang Wang¹, Bohan Chen¹, Siqi Wang¹, Hui Yang², Yuchen Liu³, Yanru Mo³, Yun Wang¹, Bin Fang⁴, Fuchun Sun⁴, Xilun Ding¹, Chi Zhang^{5,6}, Li Wen^{1*}

Corresponding author: Li Wen, liwen@buaa.edu.cn

[†]These authors contributed equally to this work.

We appreciate your insightful comments as they help notably improve our paper's quality. We have made our maximum efforts to improve the manuscript and address your comments. Significant changes have been made in the revised manuscript. Here we provided the detailed response letter to the comments from the reviewers.

Reviewer #1:

The authors have made significant efforts and revised the manuscript properly according to previous comments. Therefore, I recommend to publish this manuscript in the Nature Communications.

Response: Thank you very much for your positive comments. We will make

persistent efforts.

Reviewer #2:

This is a revised version of a previously-submitted work. The authors have thoroughly addressed my concerns by supplementing:

(1) experimental results of the sensor's resolution, signal-to-noise ratio, and dynamic response.

(2) experimental results of the sensor's environmental sensitivity, including humidity, temperature, and electromagnetic interferences, etc.

(3) A more complex task of teaching the robot to cross the obstacle and grasp an artificial flower.

I believe the added data have further improved the quality of the paper and demonstrated the system's robustness and versatility

I support for its publication in Nature Communications.

Response: Thank you very much for your positive comments. We will make persistent efforts.

Reviewer #3:

After reading the response letter, I still feel not convinced. I have to say that there is

not much technical or fundamental advancement in this manuscript that deserves publication in Nature Communications.

Response:

Many thanks for the comments! Here we want clarify this paper's advancements regarding both technical or fundamental aspects.

First, we want to emphasize that this paper's primary **fundamental** goal is "*human interactively teach soft robots to perform skilled locomotion through multimodal flexible sensory interfaces.*". We consider this goal as very important because soft robots, such as soft continuum arms, are challenging to model and program. Non-specialists often face non-negligible obstacles when working with soft robots to achieve specific movements and perform certain tasks. Thus, the blue picture of this study to teaching soft robots to perform complex motions interactively without programming. We envision that interactive teaching may expand the practical uses of soft robots, as it allows non-specialists to operate the robot for various tasks without expert familiarity.

To date, very few studies demonstrate soft robots' teaching through human interaction. There are two primary **technical** challenges to achieving soft robotic teaching through human interaction: the process requires 1) a multimodal, versatile, and robust flexible sensing device for interactions between a soft robot and human demonstrator; and 2) a user-friendly, non-programmable teaching method to transfer a human demonstrator's instructions to the soft robots.

To solve the above two **technical** challenges, in this paper, we first developed a flexible bimodal smart skin (FBSS) with both tactile and touchless sensing by integrating a triboelectric sensor with a liquid metal sensor. A triboelectric sensor can respond to touchless stimulation through electrostatic induction, and the liquid metal sensor can respond to tactile stimulation. On this basis, the implemented FBSS can unambiguously distinguish between tactile and touchless modes in real-time.

We found that tactile sensors (based on triboelectric nanogenerators and others), as reported in previous literatures, are insufficient for interaction (particularly interactive teaching) between humans and soft robots. This is because the tactile sensors can only implement “contact teaching” of robots, which is unsuitable for soft robots. Through our massive experiments, we summarize the reasons as two-fold: 1) unlike the rigid manipulator, a soft continuum manipulator’s configuration is challenging to control explicitly for a user because of the manipulator's infinite degrees of freedom and compliant nature. 2) In practice, contact teaching makes the soft continuum robots produce passive deformation, and measuring the deformed configuration of a soft robot requires a large number of soft sensors (either embedded in or on the robot's surface) to reconstruct its three-dimensional kinematics. Given these challenges, we proposed a flexible touchless and tactile bimodal sensory interface to teach soft robots interactively. We have shown in our paper the sensory interface’s principal, implementations, and interactive teaching of the soft robotic manipulators.

Through our efforts, we showed the advantages of the teaching method in the multi-degree-freedom control of the continuum soft robot, and demonstrated the

capacity of interactively teaching complex three-dimensional locomotion of the soft continuum robot. We also show that non-specialists can also quickly master the skill of teaching soft robots via the proposed method. We envision that this user-friendly, non-programmable teaching approach could broadly expand the domains in which humans interact with and utilize soft robots.

Following the guideline of Nature communications, we reorganized the text in the required order. We have also double-checked and corrected mathematical terms throughout the main text and supplementary information. Shadings or symbols in graphs have been defined in the associated legends.

We sincerely hope that the above responses can reduce your concerns and hope you can find them satisfactory.